# WHY DP "LOCAL" SGD – FASTER CONVERGENCE IN LESS COMPOSITION WITH CLIPPING BIAS REDUCTION

## ABSTRACT

We argue to apply Differentially-Private Local Stochastic Gradient Descent (DP-LSGD), a generalization of regular DP-SGD with per-sample local iterations, to systematically improve privacy-preserving machine learning. We prove and show the following facts in this paper: a). DP-LSGD with local iterations can produce more concentrated per-sample updates and therefore enables a more efficient exploitation of the clipping budget with a better utility-privacy tradeoff; b). given the same $T$ privacy composition or per-sample update aggregation, with properly-selected local iterations, DP-LSGD can converge faster in $O(1/T)$ to a small neighborhood of (local) optimum compared to $O(1/\sqrt{T})$ in regular DP-SGD, i.e., DP-LSGD produces the same accuracy while consumes less of the privacy budget. From an empirical side, thorough experiments are provided to support our developed theory and we show DP-LSGD produces the best-known performance in various practical deep learning tasks: For example with an ($\epsilon = 4, \delta = 10^{-5}$)-DP guarantee, we successfully train ResNet20 from scratch with test accuracy $74.1\%, 86.5\%$ and $91.7\%$ on CIFAR10, SVHN and EMNIST, respectively. Our code is released in an anonymous GitHub link [1].

## 1 INTRODUCTION

*Local Stochastic Gradient Descent (LSGD)* Stich (2019) and *Differential Privacy (DP)* Dwork et al. (2006); Cormode et al. (2018); Geyer et al. (2017) are two widely-used frameworks that address the issues of communication efficiency and data privacy, respectively. Rooted in the *FedAvg* framework proposed in Konečný et al. (2016), LSGD reduces the communication burden by randomly sampling participants to perform gradient descent on their local data in parallel, only aggregating updates periodically instead of at each iteration. Although LSGD is a straightforward extension of SGD in a distributed setting with lower synchronization frequency, it has empirically demonstrated strong performance in both communication efficiency and convergence rate Lin et al. (2020). When each user holds i.i.d. data, LSGD achieves provable linear speedup proportional to the number of users and asymptotic improvements in communication overhead compared to traditional distributed SGD, while maintaining comparable accuracy Khaled et al. (2020).

In the privacy preservation regime, DP Dwork et al. (2006) offers a rigorous approach to quantifying data leakage from any computation. At a high level, DP provides input-independent guarantees that ensure an adversary cannot easily infer the participation of any individual datapoint from the release. For example, classic ($\epsilon, \delta$)-DP, with small parameters $\epsilon$ and $\delta$, implies significant Type I or Type II errors in adversarial hypothesis tests aimed at guessing whether a specific individual was involved in the process Dong et al. (2022). To produce required privacy guarantees, a core challenge in DP research is determining the *sensitivity*, i.e., the maximum possible change in the output due to replacing one individual in the input set. Once the sensitivity is provided, randomization techniques, such as the Gaussian or Laplace mechanisms Dwork et al. (2014), can be accordingly applied to obfuscate the leakage or release. However, computing sensitivity is generally NP-hard Xiao & Tao (2008). Consequently, a practical alternative is the *decompose-then-compose* framework: a complex process is (approximately) decomposed into simpler subroutines, each with controllable sensitivity. A *white-box* adversary is then assumed, who observes the intermediate computations, and the overall privacy loss is bounded by composing the leakage across steps.

---

[1] https://anonymous.4open.science/r/DP-LSGD-6710/README.md

In machine learning applications, where the output of the process is a model trained on potentially sensitive data, DP-SGD Abadi et al. (2016b); Song et al. (2013) is arguably the most widely-used DP technique. As an example of the decompose-then-compose framework, DP-SGD views SGD as a sequence of adaptive gradient mean estimations. To enforce a bounded sensitivity, per-sample gradients are clipped—usually in the $l_2$-norm Abadi et al. (2016b)—to a constant $c$, which corresponds to a projection within an $l_2$-norm ball of radius $c$. Noise, determined by both the clipping threshold $c$ and the number of compositions (model updates) $T$, is then added to the clipped gradients during each iteration to ensure the privacy parameters $(\epsilon, \delta)$ under $T$-fold composition. Larger dimensions and longer convergence times $T$ (leading to more leakage) require larger noise to maintain privacy. Although DP-SGD imposes no additional assumptions on either the model or the training data, it is notorious for its utility loss, particularly in deep learning. Moreover, the bias introduced by clipping is poorly understood; it is known that, even without added noise, clipped SGD does not converge in general Chen et al. (2020).

Since DP-SGD is assumed to release intermediate per-sample aggregates, there is no essential difference between the privacy analyses of centralized and local SGD. However, in the distributed setting, alternative DP metrics like Local DP (LDP) Cormode et al. (2018) or client-level DP Geyer et al. (2017) may be applied to protect each user's local data. Interestingly, there are several connections between federated learning and DP-SGD worth noting: First, DP-SGD is a special case of DP-LSGD. DP-SGD can be viewed as involving $n$ nodes, each holding a sample, with a *virtual* server collecting clipped stochastic gradients from sampled nodes at *each* iteration and releasing a noisy gradient descent update. Similarly, DP-LSGD aggregates a subset of local gradients, clips them, but privately synchronizes updates *periodically* rather than after each iteration. The reduced communication overhead in federated learning—through less frequent synchronization in LSGD—also implies reduced leakage and a smaller composition of privacy loss. Second, the study of utility loss due to perturbation and clipping in DP-SGD is relevant to federated learning with compressed communication Basu et al. (2019), where quantization errors in the broadcasted local updates are analogous to the bias due to clipping.

Given the fundamental connections between (a) communication efficiency and privacy composition and (b) quantization/compression error and clipping bias, we are motivated to systematically improve DP-SGD from a *virtual* federated learning perspective. However, before developing useful theoretical insights, several technical challenges must be addressed.

**Utility of Released Iterate Only**: Many existing convergence results Khaled et al. (2020); Yu et al. (2019); Wang & Joshi (2021); Haddadpour & Mahdavi (2019); Woodworth et al. (2020) on non-private LSGD are developed on the (weighted) average of all iterates. These include the intermediate iterates produced during the local updates from each user or datapoint, which will not be exposed or shared. To properly characterize the effect of noise perturbation and bias, we want more fine-grained convergence analysis to measure the performance of released iterates *only*, which is also necessary for DP-LSGD: The utility of concern is only with respect to the released outputs, and anything assumed to be published would incur privacy loss and increase the scale of noise for DP guarantees.

**Clipping Bias and Data Heterogeneity**: In practice, tight sensitivity of many data processing algorithms is *intractable* and thus a very popular but artificial control is clipping. However, clipping could also bring heavy bias. In general, there is *no* convergence guarantee for clipped SGD if we *only* assume the stochastic gradient is of bounded variance Chen et al. (2020); Koloskova et al. (2023), though under more restrictive assumptions, for example, when the stochastic gradient is in a symmetric Chen et al. (2020) or light-tailed Fang et al. (2023) distribution, or provided generalized smoothness Yang et al. (2022), some (near) convergence results are known. A concise characterization of such clipping bias still largely remains open and the bias is even more complicated in the more general DP-LSGD. To provide meaningful theory to instruct bias reduction, we do *not* want to assume Lipschitz continuity or bounded gradient, which may make the analysis trivial or impractical. The desired analysis should capture the scenario with arbitrary data heterogeneity, and the results should not require a bounded difference among the local updates.

In this paper, through tackling the above-mentioned challenges, we aim to provide useful and intuitive theory to understand perturbed optimization with DP guarantees. In particular, we explain how DP-LSGD outperforms regular DP-SGD from two perspectives: a) faster convergence in less privacy composition, and b) higher clipping efficiency. Our contributions are summarized below.

**Contribution 1: Meaningful and Verifiable Assumptions.** For both convex and non-convex optimization, our presented convergence analyses for clipped DP-(L)SGD mostly require mild assumptions that the stochastic gradients/local updates are of bounded second moment (Assumption 1), rather than globally bounded gradients assumed in prior works. We ensure the parameters capturing the statistics of local updates in our assumptions and theorems are simulatable for practical learning tasks (Section 4), which forms the foundation to develop meaningful and explainable theory that can instruct systematic improvement.

**Contribution 2: Tighter Convergence Analysis**. We present the convergence analysis on the released-only noisy iterates of DP-(L)SGD for both convex and non-convex smooth optimization (Theorems 1-2). We rigorously prove that with properly-selected local iteration, DP-LSGD enjoys a faster convergence rate to a small neighborhood of a global/local optimum as compared to DP-SGD given the same aggregation or privacy composition budget $T$. That is to say, to produce the same performance, DP-LSGD theoretically requires less per-sample update aggregation, and less composition ensures better privacy guarantees compared to DP-SGD in the same setup. Moreover, for convex optimization, we present the stronger *last-iterate* analysis, i.e., the performance of model parameters finally released from last iteration, for DP-(L)SGD, which, to our knowledge, is also the first last-iterate analysis *without* assuming bounded gradients.

**Contribution 3: Clipping Bias Reduction**: Based on the theory, we then show LSGD behaves as an efficient variance reduction of local update, where multiple local gradient descents with a small learning rate cancel out substantial sampling noise, and explain why DP-LSGD enables more efficient clipping with less clipping bias compared to DP-SGD. This initiates a new research direction to apply federated learning methods to systematically improve DP optimization with bias-reduced clipped update. Empirically, we also show DP-LSGD produces the best-known performance in various deep learning tasks and setups after properly selecting local iterations. For example, in training CIFAR10 from scratch, we achieve 71.3% and 66.9% test accuracy via DP-LSGD compared to 68.9% and 63.6% achieved in De et al. (2022), for $(\epsilon = 3, \delta = 10^{-5})$ and $(\epsilon = 2, \delta = 10^{-5})$ DP guarantees, respectively.

## 1.1 RELATED WORKS

**Convergence Analysis of LSGD**: Though the idea of LSGD can be traced back to earlier works Mangasarian (1995); McDonald et al. (2010), theoretical convergence analysis is more recent. For general applications with heterogeneous data, Wang et al. (2018) studied the convex case with local GD (without sampling on either users or users' local data) but under Lipschitz continuity. Khaled et al. (2020) presented more generic and tighter analysis for LSGD without assumptions on bounded gradient for both strongly and general convex optimization. Further generalization of LSGD to the decentralized setup under arbitrary network topology was considered in Wang & Joshi (2021); Hsieh et al. (2020). However, many existing works Khaled et al. (2020); Wang & Joshi (2021); Koloskova et al. (2020) only showed the convergence rate relying on all the intermediate averages. To our knowledge, the first analysis for synchronized-only iterates was shown in Karimireddy et al. (2020). Karimireddy et al. (2020) proposed Scaffold, a generalized LSGD with careful correction on the client-drift caused by data heterogeneity. Compared to existing works, in this paper, we prove more powerful last-iterate analysis for general convex optimization with clipping and perturbation for privacy. With a different motivation, there is another line of works also studying noisy LSGD to capture the effect of compressed local updates to further save the communication cost. But, in most existing related works Basu et al. (2019); Haddadpour et al. (2021), the compression error is assumed to be independent with zero-mean. As we need to study DP-LSGD with clipped local update, which introduces bias in the local update generation, in this paper we present more involved analysis to handle such adaptive and biased perturbation.

**Convergence Analysis of DP-SGD and DP-LSGD**: Asymptotically, under Lipschitz continuity, DP-SGD is known to produce a tight utility-privacy tradeoff Bassily et al. (2014; 2019), where no bias is produced given a clipping threshold larger than the Lipschitz constant. However, without Lipschitz continuity, practical understanding of DP-SGD remains limited. On one hand, negative examples are shown in Chen et al. (2020); Zhang et al. (2022) where clipped-SGD in general will not converge with lower bounds of clipping bias shown in Koloskova et al. (2023), and in practice clipped-SGD does have a lower convergence rate, especially in deep learning applications compared to regular SGD Zhang et al. (2022). On the other hand, under more restrictive assumptions on the stochastic gradient distribution, clipped-SGD can be shown to (nearly) converge Chen et al. (2020);

Fang et al. (2023); Yang et al. (2022). A systematical characterization of the clipping bias still largely remains open. As a consequence, there is little known meaningful theory to instruct optimization algorithms with DP guarantees, and most existing private deep learning works are empirical, which aim to search for the optimal model and hyperparameters for objective training data Papernot et al. (2021); Tramer & Boneh (2021); De et al. (2022). As for DP-LSGD, to our knowledge the only known theoretical result that captures the clipping bias is Zhang et al. (2022). However, Zhang et al. (2022) assumes globally bounded gradient compared to bounded second moment as assumed in our results, and its main motivation is to study the clipping effect in client-level DP. In this paper, we show more intuitive and generic analysis of DP-LSGD for both convex and non-convex optimization, and our motivations are also very different: We set out to provide usable quantification on the utility loss due to clipping and *we argue to apply DP-LSGD both in the centralized and distributed setup*, since DP-LSGD can significantly reduce the clipping bias with a faster convergence rate.

## 2 Preliminaries

We focus on the classic Empirical Risk Minimization (ERM) problem. Given a dataset $\mathcal{D} = \{(x_i, y_i), i = 1, 2, \cdots, n\}$, the loss function is defined as $F(w) = \frac{1}{n} \cdot \sum_{i=1}^{n} f(w, x_i, y_i) = \frac{1}{n} \cdot \sum_{i=1}^{n} f_i(w)$. We will consider the cases where the loss function $f_i(w) : \mathcal{W} \to \mathbb{R}^+$ is convex or non-convex. $w^* = \arg\min_w F(w)$ represents the global optimum. Some formal definitions about the properties of the objective loss function and Differential Privacy (DP) are defined as follows.

**Definition 1** (Smoothness). *A function $f$ is $\beta$-smooth on $\mathcal{W}$ if the gradient $\nabla f(w)$ is $\beta$-Lipschitz such that for all $w, w' \in \mathcal{W}$, $\|\nabla f(w) - \nabla f(w')\| \leq \beta \|w' - w\|$.*

**Definition 2** (Convexity and Strong Convexity). *A function $f(w)$ is $\lambda$-convex on $\mathcal{W}$ if for all $w, w' \in \mathcal{W}$, $\frac{\lambda}{2} \|w - w'\|^2 \leq f(w) - f(w') - \langle \nabla f(w'), w - w' \rangle$. We call $f(w)$ general convex if $\lambda = 0$, and $f(w)$ is strongly convex if $\lambda > 0$.*

**Definition 3** (Differential Privacy Dwork et al. (2006)). *Given a universe $\mathcal{X}^*$, we say that two datasets $X, X' \subseteq \mathcal{X}^*$ are adjacent, denoted as $X \sim X'$, if $X = X' \cup x$ or $X' = X \cup x$ for some additional datapoint $x \in \mathcal{X}$. A randomized algorithm $\mathcal{M}$ is said to be $(\epsilon, \delta)$-differentially-private (DP) if for any pair of adjacent datasets $X, X'$ and any event set $O$ in the output domain of $\mathcal{M}$, it holds that*

$$\mathbb{P}(\mathcal{M}(X) \in O) \leq e^\epsilon \cdot \mathbb{P}(\mathcal{M}(X') \in O) + \delta. \tag{1}$$

With the preparation, we can now formally describe DP-(L)SGD, as Algorithm 1. The whole process of Algorithm 1 is formed of $T$ phases. In each phase, by $q$-Poisson sampling, in expectation $(nq)$ many datapoints will be selected, and we perform $K$ local gradient descents on each data point before privately aggregating their local updates. In (3), a clipping operation on a vector $v$ with threshold $c$ is defined as $\mathcal{CP}(v, c) = v \cdot \min\{1, c/\|v\|\}$, which ensures a bounded sensitivity up to $c$. Using the clipped local update (3), by selecting $e^{(t)}$ to be proper DP noise, Algorithm 1 captures DP-SGD when $K = 1$ and DP-LSGD for general $K \geq 1$ where DP-LSGD (SGD) is essentially an LSGD (SGD) with clipped local update (per-sample gradient) and additional DP noise. The privacy analysis and the noise bound are *identical* for both DP-LSGD and DP-SGD given the same clipping threshold $c$ and $T$ composition: In both methods, the aggregation of per-sample updates at the end of each phase whose sensitivity is $c$ in $l_2$-norm are privately released. Therefore, we may follow the standard composition analysis in Abadi et al. (2016a) to select noise for required $(\epsilon, \delta)$ DP guarantees.

**Lemma 1** (Abadi et al. (2016a)). *For given $(\epsilon, \delta)$ DP budget and the number of composition/phase $T$, there exist some constants $\alpha_1$ and $\alpha_2$ such that once $q \geq \alpha_1 \cdot \sqrt{\epsilon/T}$, Algorithm 1 satisfies $(\epsilon, \delta)$ DP once the noise variance $\sigma^2$ of the Gaussian noise $Q^{(t)} \sim \mathcal{N}(0, \sigma^2 \cdot \boldsymbol{I})$, $t = 1, 2, ..., T$, satisfies*

$$\sigma \geq \alpha_2 \cdot \frac{q\sqrt{T \log(1/\delta)}}{\epsilon}.$$

## 3 Utility and Clipping Bias of DP-(L)SGD

In this section, we present the convergence analysis of DP-LSGD with clipped local update (3) in Algorithm 1 and a comparison with DP-SGD. To capture incurred clipping bias, we need to introduce a new term, termed *incremental norm*.

---

**Algorithm 1** Differentially-Private Local SGD (DP-LSGD) with Noisy Clipped Periodic Averaging

1: **Input:** A dataset $\mathsf{U} = \{u_1, u_2, \cdots, u_n\}$ of $n$ datapoints, per-sample loss function $f_i(w) = \mathcal{L}(w, u_i)$, sampling rate $q$, step size $\eta$, local update length $K$ and released aggregation number $T$, clipping threshold $c$, initialization $\bar{w}^{(0)}$, and Gaussian noises $Q^{(1:T)}$ i.i.d. in $\mathcal{N}(0, \sigma^2 \cdot \boldsymbol{I})$.

2: **for** $t = 1, 2, \cdots, T$ **do**

3:     Implement i.i.d. sampling of rate $q$ to select an index batch $S^{(t)} = \big\{[1], \cdots, [B_t]\big\}$ from $\{1, 2, \cdots, n\}$ of size $B_t$.

4:     **for** $i = 1, 2, \cdots, B_t$ in parallel **do**

5:         $w_{[i]}^{(t,0)} = \bar{w}^{(t-1)}$.

6:         **for** $k = 1, 2, \cdots, K$ **do**

6:

$$w_{[i]}^{(t,k)} = w_{[i]}^{(t,k-1)} - \eta \nabla f_{[i]}(w_{[i]}^{(t,k-1)}). \tag{2}$$

7:         **end for**

8:         Clip the per-sample update in $l_2$-norm up to $c$ as

$$\Delta w_{[i]}^{(t)} = \mathcal{CP}(w_{[i]}^{(t,K)} - \bar{w}^{(t-1)}, c) = (w_{[i]}^{(t,K)} - \bar{w}^{(t-1)}) \cdot \min\{1, \frac{c}{\|w_{[i]}^{(t,K)} - \bar{w}^{(t-1)}\|_2}\}$$

9:     **end for**

10:

$$\bar{w}^{(t)} = \bar{w}^{(t-1)} + \frac{1}{nq} \cdot \Big(\sum_{i=1}^{B_t} \Delta w_{[i]}^{(t)} + Q^{(t)}\Big) \tag{3}$$

11: **end for**

12: **Output**: $\bar{w}^{(t)}$ for $t = 1, 2, \cdots, T$.

---

**Definition 4** (Incremental Norm). *In the $t$-th phase of Algorithm 1, we define $\Psi_i^{(t)} = \mathbf{1}\big(\|\Delta w_i^{(t)}\| > c\big) \cdot (\|\Delta w_i^{(t)}\| - c)$ as the incremental norm of the local update from $f_i(w)$ compared to the clipping threshold $c$, for $t = 1, 2, \cdots, T$.*

In Definition 4, the incremental norm $\Psi_i^{(t)}$ simply quantifies the difference between the norm of the local update and its clipped version from $f_i(w)$. Clearly, when the update $\Delta w_i^{(t)}$ is of bounded second moment, the second moment of its incremental norm $\Psi_i^{(t)}$ is also bounded. It is not hard to observe that the clipped local update is essentially a scaled version of the original update, and thus virtually one may view DP-LSGD as a generalization of noisy LSGD but each local update applies a different and adaptively-selected learning rate. To characterize the difference among those learning rates, we need the following assumption on the bounded-variance stochastic gradient and update.

**Assumption 1** (Bounded Variance of Stochastic Gradient). *For any $w \in \mathcal{W}$ and an index $i$ that is randomly selected from $\{1, 2, \cdots, n\}$, there exists $\tau > 0$ such that $\mathbb{E}[\|\nabla F(w) - \nabla f_i(w)\|^2] \leq \tau$.*

**Definition 5** (Second Moment Bound of Incremental Norm). *In Algorithm 1, given the selections of $K$ and $\eta$, via (3) on an objective function $F(w) = \frac{1}{n} \cdot f_i(w)$, $\mathbb{E}\big[\big(\sum_{i=1}^n (\Psi_i^{(t)})^2\big)/n\big]$ is upper bounded by $\mathcal{B}^2(K, \eta)$, for $t = 1, 2, \cdots, T$.*

The expectations in Assumption 1 and Definition 5 are both took upon the entire randomness of Algorithm 1. Definition 5 introduces a function $\mathcal{B}^2(K, \eta)$ as the upper bound of the square of $l_2$-norm of each local update. We will study $\mathcal{B}^2(K, \eta)$ later in Table 2 and 3 in practical deep learning tasks. For notation brevity, we simply use $\mathcal{B}$ to denote $\mathcal{B}(K, \eta)$ in the following. Moreover, we assume that the dimensionality of the model parameter $w$ is $d$, and thus the DP noise injected $\mathbb{E}[\|Q^{(t)}\|^2] = \sigma^2 d$.

### 3.1 UTILITY OF DP-LSGD IN CONVEX OPTIMIZATION

Another assumption we need for the analyses of DP-LSGD on general convex optimization is the similarity between $f_i$.

**Assumption 2** ($\gamma$ Similarity). *For $F(w) = 1/n \cdot \sum_{i=1}^n f_i(w)$, local functions $f_i$ are of $\gamma$-similarity to $F$ such that for any $w \in \mathcal{W}$, $|f_i(w) - F(w)| \leq \gamma$, for some constant $\gamma > 0$.*

The main reason why we need this additional Assumption 2 is because we do not assume Lipschitz continuity of $F(w)$ and we alternatively use the similarity among local functions to characterize the deviation of the evaluation of convex $F(\cdot)$ on biased iterates.

**Theorem 1** (Last-iterate of DP-LSGD in General Convex Optimization). *For an arbitrary objective function $F(w) = \frac{1}{n} \cdot \sum_{i=1}^{n} f_i(w)$ where $f_i(w)$ is convex and $\beta$-smooth, and under Assumptions 1 and 2, when $\eta = O(1/\sqrt{TK})$ and $Q^{(t)}$ is independent DP noise such that $\mathbb{E}[Q^{(t)}] = 0$ and $\mathbb{E}[\|Q^{(t)}\|^2] = \sigma^2 d$, $t = 1, 2, \cdots, T$, then when $K^2 = O(nq)$, for $(\epsilon, \delta)$-DP with $\sigma = \tilde{O}(\frac{c\sqrt{T \log(1/\delta)}}{n\epsilon})$, DP-LSGD with clipping threshold $c$ ensures that*

$$\mathbb{E}[F(\bar{w}^{(T)}) - F(w^*)] =$$

$$\tilde{O}\Big(\underbrace{\frac{c+\mathcal{B}}{c} \cdot \Big(\frac{\|\bar{w}^{(0)} - w^*\|^2}{\sqrt{TK}} + \Big(\frac{1}{\sqrt{TK}} + \frac{K}{T}\Big)\tau\Big)}_{(A)} + \underbrace{\frac{\gamma\mathcal{B}}{c}}_{(B)} + \underbrace{\frac{c+\mathcal{B}}{c} \cdot \frac{T^{3/2}K^{1/2}\log(1/\delta)dc^2}{n^2\epsilon^2}}_{(C)}\Big). \quad (4)$$

The proof can be found in Appendix D. We give a sketch here. There are three main challenges to derive the last-iterate convergence of LSGD with unbounded gradients:

i).To derive the last-iterate guarantee, we need to keep track of the progress of $F(\bar{w}^{(t)}) - F(\bar{w}^{(t')})$ for different $t$ and $t'$. To support this, we adopt the idea from Khaled et al. (2020); Zhou & Cong (2017) to consider a virtual sequence determined by the average of all intermediate updates assuming all users participate in the $t$-th phase, i.e., $\tilde{w}^{(t,k)} = \frac{1}{n} \cdot \sum_{i=1}^{n} w_i^{(t,k)}$. But instead, we show a more generic analysis on $F(\tilde{w}^{(t,k)}) - F(u)$ for arbitrary $u$ and a careful characterization of the difference between $F(\tilde{w}^{(t,k)})$ and $F(\bar{w}^{(t)})$ under sampling, given that $\bar{w}^{(t)}$ is the actual and only release.

ii). A more challenging problem is that we cannot straightforwardly apply classic last-iterate convergence analyses Zhang (2004); Shamir & Zhang (2013); Li & Orabona (2019) which must count on the assumption of bounded gradient. To address this, in the proof, we alternatively use the following two kinds of upper bounds on the gradient norm $\|\nabla F(w)\|^2 = \|\nabla F(w) - \nabla F(w^*)\|^2 \le \min\{\beta^2\|w - w^*\|^2, 2\beta(F(w) - F(w^*))\}$, which is based on the property of smoothness and convexity. With a careful analysis on $\|\tilde{w}^{(t,k)} - w^*\|^2$ for any $t$ and $k$, we propose a more generic last-iterate framework to handle unbounded and heterogeneous local update, simultaneously.

iii). To characterize the clipping bias, at a high level, clipping can be viewed that we introduce a different step size for the local iterations and the per-sample updates produced are scaled differently. We thus carefully apply the incremental norm (Definition 4) to bound the scaling difference, which then provides an upper bound of the incurred clipping bias.

Back to the theorem interpretation, we want to mention $\mathcal{B}$ in (4) is a general variable and we focus on a practical scenario where $\mathcal{B} = O(c)$, i.e., the incremental norm of local updates is in the same order of the clipping threshold $c$ selected (matched Table 3-4), and thus $(c + \mathcal{B})/c = O(1)$. From Theorem 1, we show the last-iterate utility of DP-LSGD is captured by three terms: (A) a similar convergence rate as regular LSGD, (B) a clipping bias, and (C) the DP noise variance. First, ignoring the bias and noise, DP-LSGD still enjoys a convergence rate $\tilde{O}(\frac{\|\bar{w}^{(0)} - w^*\|^2}{\sqrt{TK}} + (\frac{1}{\sqrt{TK}} + \frac{K}{T})\tau)$. Second, the clipping bias is captured by $(\gamma\mathcal{B})/c$. This matches our intuition that a larger incremental norm $\mathcal{B}$ combined with a smaller clipping threshold $c$ will imply a more significant change to the local update and thus a larger bias. The last accumulated perturbation term is determined by noise in a scale $\tilde{O}(\frac{T^{3/2}K^{1/2}\log(1/\delta)dc^2}{n^2\epsilon^2})$ injected across each phase for $(\epsilon, \delta)$-DP under $T$-fold composition.

As we consider the very generic setup with non-trivial clipping, Theorem 4 cannot be directly compared to the classic DP-utility tradeoff Bassily et al. (2014) given Lipschitz continuity, where a utility loss $\tilde{\Theta}(\sqrt{d}/n\epsilon)$ is tight for convex optimization under $(\epsilon, \delta)$-DP. However, we have the following interesting observations. First, when we take the clipping threshold $c = O(\eta) = O(1/\sqrt{TK})$ and $K = O(T \cdot d/(n^2\epsilon^2))$, DP-LSGD achieves the same optimal rate $\tilde{O}(\sqrt{d}/n\epsilon)$ Bassily et al. (2019) ignoring the clipping bias. Second and more important, when the stochastic gradient variance $\tau$ is in the same order of the clipping bias $O(\gamma\mathcal{B}/c)$, then by selecting $c = \Theta(\eta)$ and $K = \Theta(T)$, Theorem 1 suggests that DP-LSGD will converge in $O(1/T)$ to an $O(\gamma\mathcal{B}/c + \frac{d}{n^2\epsilon^2})$ neighborhood of the global optimum. As a comparison, when we select $K = 1$ in Theorem 1, it becomes the analysis of DP-SGD

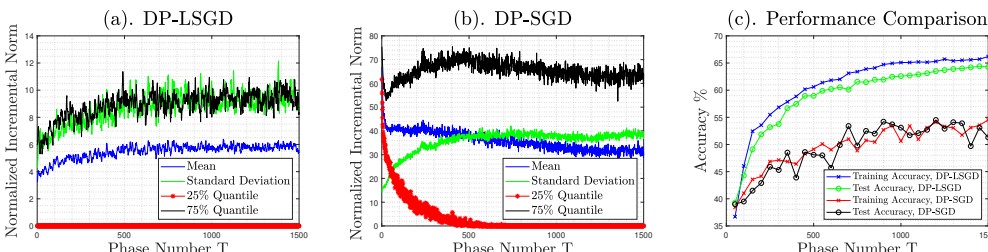

Figure 1: Training ResNet 20 on CIFAR10 with DP-LSGD ($K = 10, \eta = 0.025, c = 1$) and DP-SGD ($K = 1, \eta = 1, c = 1$) under ($\epsilon = 2, \delta = 10^{-5}$)-DP, with expected batch size 1000.

but the convergence rate to the neighborhood of global optimum in the same scale $O(\gamma \mathcal{B}/c + \frac{d}{n^2\epsilon^2})$ is only $O(1/\sqrt{T})$. Moreover, as we will show in the next section, the local update bound $\mathcal{B}$ in DP-SGD with $K = 1$ in practice would be much larger than that of DP-LSGD with a relatively larger $K$. As a simple generalization, we also include an analysis of DP-LSGD on strongly-convex functions in Appendix E, and we move our focus to the non-convex optimization in the following.

## 3.2 UTILITY OF DP-LSGD IN NON-CONVEX OPTIMIZATION

**Theorem 2** (DP-LSGD in Non-convex Optimization). *For $F(w) = \frac{1}{n} \cdot \sum_{i=1}^n f_i(w)$ where $f_i(w)$ is $\beta$-smooth and satisfies Assumption 1, when $\eta = O(1/K)$, DP-LSGD ensures that*

$$\mathbb{E}\left[\frac{\sum_{t=1}^T \|\nabla F(\bar{w}^{(t-1)})\|^2}{T}\right] \leq \frac{4F(\bar{w}^{(0)})}{TK\eta} + \frac{16\eta^2\tau\beta^2K^2}{nq} + \frac{4(1+\beta\eta)(\mathcal{B}^2/q + \sigma^2 d)}{\eta^2 K}.$$

*When we select $\eta = O(\frac{1}{\sqrt{TK}})$ and $K = \Theta(T)$, for $(\epsilon, \delta)$-DP we have that*

$$\mathbb{E}\left[\frac{\sum_{t=1}^T \|\nabla F(\bar{w}^{(t-1)})\|^2}{T}\right] = \tilde{O}\left(\frac{F(\bar{w}^{(0)})}{T} + \frac{\tau}{nq} + \frac{\mathcal{B}^2 T}{q} + \frac{d}{n^2\epsilon^2}\right). \tag{5}$$

The proof can be found in Appendix F. For the analysis of DP-LSGD in non-convex optimization, we do *not* need Assumption 2 on the similarity among local functions. To have a clearer picture, we similarly consider a practical scenario when $\mathcal{B} = \mathcal{B}_0 \cdot \eta$ for some constant $\mathcal{B}_0$ and the variance $\tau$ is also some constant. Then, with $\eta = O(\frac{1}{\sqrt{TK}})$ and $\mathcal{B}^2 = O(\frac{\mathcal{B}_0^2}{TK})$, from (5) we have that

$$\mathbb{E}\left[\frac{\sum_{t=1}^T \|\nabla F(\bar{w}^{(t-1)})\|^2}{T}\right] = O\left(\frac{F(\bar{w}^{(0)})}{T} + \frac{1}{nq} + \frac{\mathcal{B}_0^2}{qK} + \frac{d}{n^2\epsilon^2}\right) = \tilde{O}\left(\frac{1}{T} + \frac{1}{qK} + \frac{d}{n^2\epsilon^2}\right).$$

In other words, similar to the convex case, DP-LSGD will converge at a rate of $O(1/T)$ to an $\tilde{O}(1 + d/(n^2\epsilon^2))$ neighborhood of an optimum given some constant sampling rate $q$. As a comparison, for DP-SGD when $K = 1$, from Theorem 2, we can only ensure an $O(1/\sqrt{T})$ convergence rate to a same $\tilde{O}(1 + d/(n^2\epsilon^2))$ neighborhood.

**Remark 1.** *As a final remark, of independent interests, our theory can be further generalized to study the convergence rate of LSGD with more general and possibly biased perturbation, where clipping error with DP noise studied above is a special case. We defer those results to Appendix A.*

## 4 CLIPPING BIAS REDUCTION IN DP-LSGD

Throughout the previous section, we demonstrated that, asymptotically, given the same composition budget $T$, DP-LSGD achieves a faster convergence rate to a neighborhood of the (global/local) optimum compared to DP-SGD. We characterized the clipping bias primarily in terms of the second moment upper bound $\mathcal{B}^2$ of the incremental norm $\Psi_i^{(t)}$ of the local updates. In this section, we proceed to empirically analyze $\Psi_i^{(t)}$ and explore the tradeoff between clipping bias and DP (Gaussian) noise in practical deep learning tasks. We will also explain why DP-LSGD induces a smaller bias

| Dataset and Method \ $\epsilon$ | 1.0 | 1.5 | 2.0 | 2.5 | 3.0 | 4.0 |
|---|---|---|---|---|---|---|
| CIFAR10, DP-LSGD ($K = 10$) | 56.5($\pm$0.3) | 59.4($\pm$0.5) | 64.0($\pm$0.3) | 66.2($\pm$0.4) | 67.7($\pm$0.3) | 71.3($\pm$0.3) |
| CIFAR10, DP-SGD ($K = 1$) | 37.6($\pm$1.7) | 49.8($\pm$1.2) | 58.7($\pm$1.0) | 59.9($\pm$1.2) | 60.6($\pm$0.8) | 64.8($\pm$0.6) |
| SVHN, DP-LSGD ($K = 10$) | 62.4($\pm$0.9) | 83.2($\pm$0.4) | 84.4($\pm$0.5) | 85.7($\pm$0.5) | 85.4($\pm$0.4) | 86.5($\pm$0.3) |
| SVHN, DP-SGD ($K = 1$) | 55.9($\pm$1.1) | 74.5($\pm$0.8) | 78.2($\pm$0.6) | 79.8($\pm$0.6) | 80.3($\pm$1.0) | 82.2($\pm$0.5) |
| EMNIST, DP-LSGD ($K = 10$) | 89.7($\pm$0.6) | 90.2($\pm$0.4) | 90.6($\pm$0.3) | 90.9($\pm$0.3) | 91.3($\pm$0.3) | 91.7($\pm$0.3) |
| EMNIST, DP-SGD ($K = 1$) | 88.1($\pm$0.5) | 89.2($\pm$0.5) | 89.8($\pm$0.3) | 90.3($\pm$0.4) | 90.5($\pm$0.2) | 91.0($\pm$0.2) |

Table 1: **Test Accuracy** of ResNet20 on CIFAR10, SVHN, and EMNIST via DP-LSGD and DP-SGD Abadi et al. (2016b); Dörmann et al. (2021) under various $\epsilon$ and fixed $\delta = 10^{-5}$, with expected batch size 1000. Each subcase takes 5 independent trials.

and enables more efficient clipping compared to DP-SGD. All experiments were conducted using eight NVIDIA A100 (80G) GPUs.

To achieve an optimal utility-privacy tradeoff, proper selection of the clipping threshold $c$ is crucial. Previous works have focused on optimizing $c$ through either grid search Tramer & Boneh (2021) or adaptive fine-tuning Andrew et al. (2021). A smaller $c$ requires less DP noise. However, as shown in Theorems 1 and 2, a smaller $c$, and consequently a larger $\mathcal{B}$, will lead to greater clipping bias. Thus, from a signal-to-noise ratio (SNR) perspective, an ideal scenario is for the $l_2$-norm of each local update to be concentrated, allowing the clipping threshold $c$ to be maximally efficient with minimal clipping effect on most updates. Based on our theory of clipping bias, we expect that, for a given $c$, the incremental norm $\Psi_i^{(t)}$ remains small, as captured by $\mathcal{B}$ in (4) and (5).

In Fig. 1 (a,b), we plot various statistics of the incremental norm $\Psi_i^{(t)}$ for DP-LSGD and DP-SGD, respectively, on the CIFAR10 training dataset Krizhevsky et al. (2009). As per our analysis, DP-LSGD generally employs a smaller learning rate $\eta$. To ensure a fair comparison, we consider the normalized incremental norm $\Psi_i^{(t)}/\eta$. Given the same clipping threshold, comparing Fig. 1 (a) and (b), the mean of the normalized incremental norm (blue line), corresponding to $\mathcal{B}_0 = \mathcal{B}/\eta$ in our theorems, is approximately 15.2% of that for DP-SGD. The corresponding standard deviation is only around 24.3% of that of DP-SGD. A comparison of the 25% and 75% quantiles further suggests that a greater proportion of local updates experience less clipping under DP-LSGD, resulting in higher clipping efficiency. Similar observations are reported for ResNet20 training on SVHN Netzer et al. (2011) in Fig. 4 (Appendix G).

In Fig. 1 (c), we compare the performance of DP-LSGD and DP-SGD, which aligns with our theory that DP-LSGD exhibits a smaller clipping bias and a faster convergence rate. The smaller incremental norm in DP-LSGD is expected: with a relatively larger $K$, though the $K$ local gradients for each individual function $f_i(w)$ are correlated (since they derive from a single sample), their aggregation averages out a significant amount of sampling noise, leading to more concentrated $l_2$-norms for local updates. Table 1 presents additional comparisons of their performance on CIFAR10 Krizhevsky et al. (2009), SVHN Netzer et al. (2011), and EMNIST Cohen et al. (2017). Hyperparameters, such as the number of iterations $T$ and learning rate $\eta$, were fine-tuned for both DP-SGD and DP-LSGD, as detailed in Section G. The DP-SGD performance in Table 1 also matches that of previous works, such as ResNet20 results on CIFAR10 Dörmann et al. (2021), which report 58.6% test accuracy at ($\epsilon = 1.96, \delta = 10^{-5}$) and 66.2% at ($\epsilon = 4.2, \delta = 10^{-5}$).

We also compare DP-LSGD with the state-of-the-art results in De et al. (2022), which suggest that larger batch sizes significantly improve the utility-privacy tradeoff. In Table 2, we apply this idea by scaling the batch size from 1,000 (as used in Table 1) to 8,192, incorporating other advanced techniques such as weight standardization and parameter averaging from De et al. (2022). We compare DP-LSGD with DP-SGD on both ResNet20 and WideResNet-40-4, showing that DP-LSGD can also benefit from these improvements and outperforms De et al. (2022), particularly in high- and medium-privacy regimes, while in lower-privacy regimes larger neural network with stronger learning capacity is advantageous.

Finally, in Figure 2, we compare the corresponding clipping bias bound $B(K, \eta)/c$ from Theorem 1, averaged across the initial 100 phases of DP-(L)SGD. The clipping threshold takes the form $c = 25 \cdot K\eta$, where $c$ scales with $K$ and $\eta$. The constant factor of 25 was empirically determined to yield optimal performance on the CIFAR10 dataset. In Figure 2, we present the ratio $B(K, \eta)/c(K, \eta)$,

| Method \ $\epsilon$ | 1.0 | 2.0 | 3.0 | 4.0 | 5.0 | 6.0 | 7.0 | 8.0 |
|---|---|---|---|---|---|---|---|---|
| DP-LSGD on ResNet 20 | **59.2** | **66.9** | **71.3** | **74.1** | 74.8 | 75.5 | 76.4 | 77.9 |
| DP-LSGD on WideResNet 40-4 | 57.0 | 64.7 | 70.2 | 73.4 | **75.1** | **78.6** | **79.4** | **80.6** |
| DP-SGD on WideResNet 40-4 De et al. (2022) | 53.4 | 63.6 | 68.9 | 72.5 | 74.3 | 77.8 | 79.0 | 80.3 |

Table 2: **Test Accuracy** of ResNet20 and WRN-40-4 on CIFAR10 via DP-LSGD with expected larger batch size 8,192 and DP-SGD on WRN-40-4 with batch size 8,192 (reproduction) De et al. (2022) under various $\epsilon$ and fixed $\delta = 10^{-5}$.

| $K \setminus \eta$ | 0.01 | 0.02 | 0.03 | 0.04 | 0.05 |
|---|---|---|---|---|---|
| $K = 1$ | 1.74 | 1.73 | 1.72 | 1.73 | 1.72 |
| $K = 4$ | 1.33 | 1.30 | 1.28 | 1.26 | 1.24 |
| $K = 8$ | 0.92 | 0.87 | 0.83 | 0.82 | 0.78 |
| $K = 12$ | 0.60 | 0.54 | 0.49 | 0.47 | 0.46 |
| $K = 16$ | 0.36 | 0.31 | 0.27 | 0.25 | 0.25 |
| $K = 20$ | 0.18 | 0.14 | 0.11 | 0.13 | 0.12 |

| $K \setminus \eta$ | 0.01 | 0.02 | 0.03 | 0.04 | 0.05 |
|---|---|---|---|---|---|
| $K = 1$ | 0.68 | 1.32 | 2.06 | 2.70 | 3.40 |
| $K = 4$ | 2.33 | 4.63 | 6.82 | 9.07 | 11.2 |
| $K = 8$ | 3.81 | 7.43 | 10.93 | 14.36 | 17.8 |
| $K = 12$ | 4.74 | 9.12 | 13.21 | 17.23 | 21.34 |
| $K = 16$ | 5.34 | 10.10 | 14.48 | 18.71 | 23.28 |
| $K = 20$ | 5.67 | 10.61 | 15.08 | 19.55 | 24.09 |

Figure 2: The Ratio between **Incremental Norm** and **Clipping Threshold** $B(K)/c(K, \eta)$ of DP-(L)SGD on CIFAR10.

Figure 3: **Average** $l_2$-**Norm** of Local Update from DP-(L)SGD when training ResNet20 over CIFAR10 over the Initial 100 Phases.

which captures the bound on the clipping bias from Theorem 1. As expected, larger values of $K$ lead to more concentrated local updates and improved clipping efficiency.

Figure 3 further showcases the average $l_2$-norm of the local updates across different combinations of local gradient descent number $K$ and step size $\eta$. On one hand, for a given stepsize (within each column), a discernible trend emerges: the rate of increase in the $l_2$ norm of the local update decelerates as $K$ escalates. This observation lends credence to our assertion that the sampling noises originating from local gradients—despite their interdependence and evaluation on the same datapoint—tend to cancel out substantially. On the other hand, when focusing on a fixed value of $K$ (within each row), it becomes evident that the norm of the local update maintains a linear proportionality with the step size $\eta$, which matches our intuition.

**Remark 2.** *We would like to comment on the discrepancy between the theoretical and practical selection of the local iteration number $K$. Theorems 1 and 2 suggest that $K$ should be selected as $K = \Theta(T)$. However, the above empirical findings indicate that in many deep learning tasks, the optimal choice of $K$ tends to be a constant. We believe there are two main reasons for this difference. First, practical neural networks do not exhibit ideal smoothness with a constant smoothness parameter, as described in Definition 1. In Figures 2 and 3, we observe that when $K > 25$, the divergence between local updates increases significantly. Second, in DP-(L)SGD, the number of iterations $T$ cannot be arbitrarily large due to privacy budget constraints, limiting the feasible range for $K$.*

## 5 CONCLUSION AND PROSPECTS

In this paper, we advanced the understanding upon the effect of local iterations on clipping bias and convergence rate in privacy-preserving gradient methods. We established the connections between the bias and the second moment of local updates and explain how DP-LSGD outperforms DP-SGD in less composition with automatic clipping bias reduction. This initializes a new direction to systematically instruct private learning by connecting the research of variance reduction in distributed optimization, where more advanced acceleration methods Karimireddy et al. (2020); Haddadpour et al. (2021); Mitra et al. (2021) in federated learning to reduce the "local-update drift" caused by data heterogeneity could be potentially applied to further improve the clipping bias given local updates of smaller variance.

**Potential Limitations and Improvements**: Compared to DP-SGD, one drawback of DP-LSGD is the relatively larger memory and timing consumption. In all above-presented experiments, we simulate DP-LSGD by computing each local update in parallel at a cost of storing the local iterates from each data point selected. For DP-SGD, many PyTorch libraries with fast per-sample gradient computation in optimized memory overhead have been developed, such as Opacus Yousefpour et al.

(2021). In addition, provided a same communication budget $T$, though DP-LSGD can converge faster with better utility compared to DP-SGD, its running time is also $K$ times longer since $K$ times more gradient computation is needed. Therefore, another promising future direction is, from software level, to design more efficient implementation of DP-LSGD.

## 6 REPRODUCIBILITY AND ETHICS STATEMENT

We release our code in an annonymous GitHub link `https://anonymous.4open.science/r/DP-LSGD-6710/README.md` and the optimized hyper-parameter selections for both DP-LSGD and DP-SGD are detailed in Appendix G.

This paper does not propose or apply any new datasets for the experiments and the authors do not see any potential ethical issues related to this paper.

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

# A CONVERGENCE OF SYNCHRONIZED-ONLY ITERATE IN LSGD UNDER GENERAL PERTURBATION

We present a generalized version of clipped DP-LSGD (Algorithm 1) with general perturbation $Q^{(1:T)}$ in iterates, as Algorithm 2. The clipping error and DP noise, considered in Algorithm 1, can be viewed as a special case.

We have two important remarks on Algorithm 1 and the simple generalization of the centralized DP model considered in Algorithm 1.

(a) **Full and Stochastic Local Gradient**: As mentioned before, for DP-(L)SGD in the centralized DP model, each $f_i$ is determined by a single datapoint, and thus the stochastic and the full gradient of $f_i$ are the same. However, Algorithm 1 and our analysis can be easily generalized to the stochastic local gradient case since we always assume a general perturbation term $Q^{(t)}$ across phases: the independent zero-mean sampling noise can be captured by $Q^{(t)}$. Besides, when we compare with existing works in the following, we always fairly compare those results in the same full local gradient setup.

(b) **A Unified Analysis of Centralized/Local/Client DP**: We also want to stress that our motivation to study DP-LSGD is not because we focus on the federated setup, but to provide a unified analysis of the clipping bias and argue for using DP-LSGD *even in the centralized setup*. Our results are straightforwardly applicable to all setups for centralized, local Cormode et al. (2018) and client-level Geyer et al. (2017) DP. The only difference is that different scales of noise, captured by $Q^{(t)}$ are required, determined by the number of all datapoints, local datapoints and the users involved, respectively.

In the following, we will study the convergence analysis of LSGD in Algorithm 2 using the non-clipped local update (7) for both convex and non-convex optimization.

**Theorem 3** (Last-iterate Convergence of Noisy LSGD in General Convex Optimization). *For an objective function $F(w) = \frac{1}{n} \cdot \sum_{i=1}^{n} f_i(w)$ where $f_i(w)$ is convex and $\beta$-smooth with variance-bounded gradient (Assumption 1), when $\eta < \min\{\frac{\beta}{\sqrt{24K}}, \frac{1}{20\beta}, \frac{1}{2\beta+3K\beta/(nq)}\}$, $\log(TK) \geq 2$, and $Q^{(t)}$ is an independent noise such that $\mathbb{E}[Q^{(t)}] = 0$ and $\mathbb{E}[\|Q^{(t)}\|^2] \leq \bar{\mathcal{Q}}$, for some parameter $\bar{\mathcal{Q}}$ for $t = 1, 2, \cdots, T$, when $K^2 = O(nq)$ and $\eta = O(1/\sqrt{TK})$, Algorithm 2 with (7) ensures*

$$\mathbb{E}[F(\bar{w}^{(T)})] = \tilde{O}(\frac{\|\bar{w}^{(0)} - w^*\|^2}{\sqrt{TK}} + \frac{\tau}{\sqrt{TK}} + \frac{K\tau}{T} + \sqrt{TK}\bar{\mathcal{Q}}).$$

The full proof can be found in Appendix B.

When there is no noise $\bar{\mathcal{Q}} = 0$, provided that $K = O(T^{1/3})$, we show LSGD achieves $\tilde{O}(\frac{\|\bar{w}^{(0)} - w^*\|^2 + \tau}{T^{2/3}})$ last-iterate convergence in general-convex optimization.

We now study the non-convex scenario. Assumption 1 is the only additional assumption we need for the analysis of non-private LSGD without clipping.

**Theorem 4** (Synchronized-only Iterate Convergence of Noisy LSGD in Non-convex Optimization). *For an arbitrary objective function $F(w) = \frac{1}{n} \cdot \sum_{i=1}^{n} f_i(w)$, where $f_i(w)$ is $\beta$-smooth and satisfies Assumption 1, and for arbitrary perturbation (not necessarily independent or of zero mean) where $\mathbb{E}[\|Q^{(t)}\|^2] \leq \bar{\mathcal{Q}}$, when $\eta < \min\{\frac{\beta}{\sqrt{24K}}, \frac{1}{4\beta K}, \frac{1}{20\beta}\}$, Algorithm 2 with (7) ensures that*

$$\mathbb{E}[\frac{\sum_{t=1}^{T}\|\nabla F(\bar{w}^{(t-1)})\|^2}{T}] = O(\frac{\tau^{1/3}}{T^{2/3}(nq)^{1/3}} + \frac{T^{2/3}\tau^{2/3}K\bar{\mathcal{Q}}}{(nq)^{2/3}}),$$

*when we select $\eta = O(\frac{(nq)^{1/3}}{T^{1/3}K\tau^{1/3}})$. In particular, when $Q^{(t)}$ is independent and $\mathbb{E}[Q^{(t)}] = 0$, and $\eta = \Theta(1/K)$, then*

$$\mathbb{E}[\frac{\sum_{t=1}^{T}\|\nabla F(\bar{w}^{(t-1)})\|^2}{T}]$$
$$\leq O(\frac{F(\bar{w}^{(0)})}{\eta TK} + \tau + \frac{\sum_{t=1}^{T}\beta\mathbb{E}[\|Q^{(t)}\|^2]}{\eta TK}) = O(\frac{1}{T} + \tau + \bar{\mathcal{Q}}).$$

---

**Algorithm 2** (Differentially Private) Local SGD with Noisy (Clipped) Periodic Averaging

---

1: **Input:** A system of $n$ workers where each holds a local loss function $F(w) = f_i(w)$, sampling rate $q$, update step size $\eta$, local update length $K$ and global synchronization number $T$, and initialization $\bar{w}^{(0)}$ with synchronization noise $Q^{(1:T)}$.

2: **for** $t = 1, 2, \cdots, T$ **do**

3:     Implement i.i.d. sampling to select an index batch $S^{(t)} = \{[1], \cdots, [B_t]\}$ from $\{1, 2, \cdots, n\}$ of size $B_t$.

4:     **for** $i = 1, 2, \cdots, B_t$ in parallel **do**

5:         $w_{[i]}^{(t,0)} = \bar{w}^{(t-1)}$.

6:         **for** $k = 1, 2, \cdots, K$ **do**

6:

$$w_{[i]}^{(t,k)} = w_{[i]}^{(t,k-1)} - \eta \nabla f_{[i]}(w_{[i]}^{(t,k-1)}). \tag{6}$$

7:         **end for**

8:     **end for**

8:

$$\bar{w}^{(t)} = \frac{1}{nq} \cdot \left(\sum_{i=1}^{B_t} w_{[i]}^{(t,K)}\right) + Q^{(t)}. \tag{7}$$

9: **end for**

10: **Output**: $\bar{w}^{(t)}$ for $t = 1, 2, \cdots, T$.

---

The proof can be found in Appendix C. In Theorem 4, we provide an analysis on the effect of generic perturbation, which can also be used to capture the clipping bias in DP-LSGD. When there is no perturbation, Theorem 4 has two implications. First, we show to ensure $\min \mathbb{E}[\|\nabla F(\bar{w}^{(t)})\|^2] \leq \kappa$, we need $T = O(\frac{\sqrt{\tau/(nq)}}{\kappa^{3/2}})$, which is tighter than the state-of-the-art results $O(\frac{\tau/(nq)}{\kappa^2} + \frac{\sqrt{\tau}}{\kappa^{3/2}})$ in Karimireddy et al. (2020). Second, compared to $O(1/T^{2/3})$, we also show that LSGD can converge faster in $O(1/T)$ to a *$\tau$-neighborhood* of the optimum. This is helpful to understand the practical performance of DP-LSGD, as discussed in Section 3.2.

As a final remark, we want to mention it is possible to improve the convergence rate from $O(1/T^{2/3})$ to $O(1/T)$ via careful variance reduction or an error feedback mechanism, such as Scaffold Karimireddy et al. (2020) or FedLin Mitra et al. (2021). However, the implementation of those advanced tricks in DP-LSGD with additional sensitivity control is not clear and in this paper we only focus on the standard LSGD.

# B  PROOF OF THEOREM 3: LAST-ITERATE CONVERGENCE OF NOISY LSGD IN GENERAL CONVEX OPTIMIZATION

We first present a sketch of the proof. There are two main challenges to derive the last-iterate convergence of LSGD with unbounded gradients. First, to derive the last-iterate guarantee, we need to keep track of the progress of $F(\bar{w}^{(t)}) - F(\bar{w}^{(t')})$ for different $t$ and $t'$. To support this, we still adopt the similar idea from existing works Khaled et al. (2020); Zhou & Cong (2017) to consider a virtual sequence determined by the average of all intermediate updates assuming all users participate in the $t$-th phase, i.e., $\tilde{w}^{(t,k)} = \frac{1}{n} \cdot \sum_{i=1}^{n} w_i^{(t,k)}$. But instead, we show a more generic analysis on $F(\tilde{w}^{(t,k)}) - F(u)$ for arbitrary $u$ and a careful characterization of the difference between $F(\tilde{w}^{(t,k)})$ and $F(\bar{w}^{(t)})$ under sampling, given that $\bar{w}^{(t)}$ is the actual and only release. The second and more challenging problem is that we cannot straightforwardly apply classic last-iterate convergence analyses Zhang (2004); Shamir & Zhang (2013); Li & Orabona (2019) which must count on the assumption of bounded gradient. To address this, in the proof, we alternatively use the following two kinds of upper bounds on the gradient norm

$$\|\nabla F(w)\|^2 = \|\nabla F(w) - \nabla F(w^*)\|^2 \leq \min\{\beta^2 \|w - w^*\|^2, 2\beta(F(w) - F(w^*))\},$$

which is based on the property of smoothness and convexity. With a careful analysis on $\|\tilde{w}^{(t,k)} - w^*\|^2$ for any $t$ and $k$, we propose a more generic last-iterate framework to handle unbounded and heterogeneous local update, simultaneously.

## B.1 MAIN PROOF

Before the start, we define a virtual sequence $\hat{w}^{(t,k)} = \bar{w}^{(t-1)} + \frac{1}{nq}\sum_{i=1}^{n} 1_i^{(t)}(w_i^{(t,k)} - \bar{w}^{(t-1)})$ for those intermediate iterates produced by the users selected in the $t$-th phase. $1_i^{(t)}$ is an indicator which equals 1 iff the $i$-th user is selected in the $t$-th phase. Meanwhile, we imagine the scenario that all users participate in the $t$-th phase computation and a sequence of intermediate iterates $w_i^{(t,k)}$ for $i = 1, 2, \cdots, n$, and $k = 1, 2, \cdots, K$, is produced. We use $\tilde{w}^{(t,k)} = \frac{1}{n} \cdot \sum_{i=1}^{n} w_i^{(t,k)}$ to denote the average. It is not hard to observe that $\mathbb{E}[\hat{w}^{(t,k)}] = \tilde{w}^{(t,k)}$ conditional on $\bar{w}^{(t-1)}$. Moreover, $w_i^{(t,0)} = \tilde{w}^{(t,0)} = \bar{w}^{(t-1)}$ for $i = 1, 2, \cdots, n$. In the following, we unravel $\|\tilde{w}^{(t,k)} - u\|^2$ for arbitrary $u$ and obtain

$$\|\hat{w}^{(t,k)} - u\|^2 = \|\hat{w}^{(t,k-1)} - \frac{\eta}{nq}\sum_{i=1}^{n} 1_i^{(t)} \nabla f_i(w_i^{(t,k-1)}) - u\|^2$$

$$= \|\hat{w}^{(t,k-1)} - u\|^2 - \frac{2}{nq} \cdot \sum_{i=1}^{n} \eta 1_i^{(t)} \cdot \langle \hat{w}^{(t,k-1)} - u, \nabla f_i(w_i^{(t,k-1)})\rangle + \|\frac{\sum_{i=1}^{n} \eta 1_i^{(t)} \nabla f_i(w_i^{(t,k-1)})}{nq}\|^2.$$

$$(8)$$

We first work on the last term $\|\frac{\sum_{i=1}^{n} \eta 1_i^{(t)} \nabla f_i(w_i^{(t,k-1)})}{nq}\|^2$ in (8).

**Lemma 2.** *Conditional on $\bar{w}^{(t-1)}$,*

$$\mathbb{E}[\|\frac{\sum_{i=1}^{n} \eta 1_i^{(t)} \nabla f_i(w_i^{(t,k-1)})}{nq}\|^2] \leq \frac{10\eta^2\beta^2}{n}\sum_{i=1}^{n}\|w_i^{(t,k-1)} - \tilde{w}^{(t,k-1)}\|^2 + \frac{6\eta^2\tau}{nq}$$

$$+ 10\eta^2 \min\{2\beta(F(\tilde{w}^{(t,k-1)}) - F(w^*)), \beta^2\|\tilde{w}^{(t,k-1)} - w^*\|^2\}.$$

$$(9)$$

Now, we move our focus to the second term $\frac{-2}{nq} \cdot \sum_{i=1}^{n} \eta 1_i^{(t)} \cdot \langle \hat{w}^{(t,k-1)} - u, \nabla f_i(w_i^{(t,k-1)})\rangle$ of (8).

**Lemma 3.** *Conditional on $\bar{w}^{(t-1)}$,*

$$\mathbb{E}\left[-\frac{2}{nq} \cdot \sum_{i=1}^{n} \eta 1_i^{(t)} \langle \hat{w}^{(t,k-1)} - u, \nabla f_i(w_i^{(t,k-1)})\rangle\right]$$

$$\leq 2\eta\left(F(u) - F(\tilde{w}_i^{(t,k-1)}) + \frac{\beta}{2n}\sum_{i=1}^{n}\|w_i^{(t,k-1)} - \tilde{w}^{(t,k-1)}\|^2\right).$$

$$(10)$$

Finally, we consider the upper bound of $\sum_{i=1}^{n}\|w_i^{(t,k-1)} - \tilde{w}^{(t,k-1)}\|^2$.

**Lemma 4.** *When $\eta < \frac{1}{\sqrt{24}\beta K}$,*

$$\sum_{i=1}^{n}\|w_i^{(t,k)} - \tilde{w}^{(t,k)}\|^2 \leq 4k^2 n\tau\eta^2.$$

$$(11)$$

Now, we combine Lemma 2, 3 and 4 together and go back to (8). On one hand, when we adopt the upper bound of Lemma 2 using $F(\tilde{w}^{(t,k)}) - F(w^*)$, we have

$$\mathbb{E}[\|\hat{w}^{(t,k)} - u\|^2] \leq \mathbb{E}\big[\|\hat{w}^{(t,k-1)} - u\|^2 + 20\eta^2\beta\big(F(\tilde{w}^{(t,k-1)}) - F(w^*)\big) + 2\eta(F(u) - F(\tilde{w}^{(t,k-1)}))$$

$$+ \frac{6\eta^2\tau}{nq} + (10\eta^2\beta^2 + \beta\eta) \cdot 4k^2\tau\eta^2\big].$$

$$(12)$$

Sum up (12) on both sides from $k = 1, 2, \cdots, K$, and we have that

$$
\mathbb{E}\big[\sum_{k=1}^{K} 2\eta(F(\tilde{w}^{(t,k-1)}) - F(u)) - 20\eta^2\beta\big(F(\tilde{w}^{(t,k-1)}) - F(w^*))\big)\big]
$$

$$
\leq \mathbb{E}[\|\bar{w}^{(t-1)} - u\|^2 - \|\hat{w}^{(t,K)} - u\|^2] + \frac{6K\eta^2\tau}{nq} + (10\eta^2\beta^2 + \beta\eta) \cdot 4K^3\tau\eta^2.
$$

(13)

When $u = w^*$, it is noted that the left side of (13) becomes

$$
\mathbb{E}\big[\sum_{k=1}^{K} (2\eta - 20\eta^2\beta)(F(\tilde{w}^{(t,k-1)}) - F(w^*))\big],
$$

and once $\eta$ is small enough such that $2(\eta - 10\eta^2\beta) > 0$ where $\eta < 1/(10\beta)$, then the above is non-negative. In the following, we further take the perturbation $Q^{(t)}$ into accountant. It is noted that

$$
\mathbb{E}[\|\bar{w}^{(t)} - u\|^2] = \mathbb{E}[\|\hat{w}^{(t,K)} + Q^{(t)} - u\|^2] = \mathbb{E}[\|\hat{w}^{(t,K)} - u\|^2] + \mathbb{E}[\|Q^{(t)}\|^2], \quad (14)
$$

since $Q^{(t)}$ is independent zero-mean noise. Therefore, when we further sum up (13) for $t = 1, 2, \cdots, T$ combined with (14),

$$
\mathbb{E}[\frac{\sum_{t=1}^{T} \sum_{k=1}^{K} F(\tilde{w}^{(t,k)}) - F(w^*)}{TK}]
$$

$$
\leq \frac{\|\bar{w}^{(0)} - w^*\|^2}{(2\eta - 20\eta^2\beta)TK} + \frac{(6\eta^2\tau/(nq) + (10\eta^2\beta^2 + \beta\eta) \cdot 4K^2\tau\eta^2) + \bar{Q}/K}{(2\eta - 20\eta^2\beta)}.
$$

(15)

Here, as assumed $\mathbb{E}[\|Q^{(t)}\|^2] \leq \bar{Q}$. When $\eta < 1/(20\beta)$, which suggests that $(2\eta - 20\eta^2\beta) \geq \eta$ and $(10\eta^2\beta^2 + \beta\eta) \leq 2\beta\eta$, respectively, (15) can be simplified as

$$
\mathbb{E}[\frac{\sum_{t=1}^{T} \sum_{k=1}^{K} F(\tilde{w}^{(t,k)}) - F(w^*)}{TK}] \leq \frac{\|\bar{w}^{(0)} - w^*\|^2}{\eta TK} + (\frac{6\eta\tau}{nq} + 8\beta K^2\tau\eta^2) + \bar{Q}/(\eta K) \quad (16)
$$

On the other hand, when we apply Lemma 2 in (12) if we adopt the form $\beta^2\|\tilde{w}^{(t,k-1)} - w^*\|^2$ as the upper bound, we have

$$
\mathbb{E}[\|\hat{w}^{(t,k)} - u\|^2] \leq \mathbb{E}\big[\|\hat{w}^{(t,k-1)} - u\|^2 + 10\eta^2\beta^2\|\tilde{w}^{(t,k-1)} - w^*\|^2 + 2\eta(F(u) - F(\tilde{w}^{(t,k-1)}))
$$

$$
+ \frac{6\eta^2\tau}{nq} + (10\eta^2\beta^2 + \beta\eta) \cdot 4k^2\tau\eta^2\big].
$$

(17)

With a similar reasoning, when $\eta < 1/(20\beta)$,

$$
\mathbb{E}[F(\tilde{w}^{(t,k-1)}) - F(u)]
$$

$$
\leq \mathbb{E}\big[\frac{\|\hat{w}^{(t,k-1)} - u\|^2 - \|\hat{w}^{(t,k)} - u\|^2}{2\eta} + 5\eta\beta^2\|\tilde{w}^{(t,k-1)} - w^*\|^2 + \frac{3\eta\tau}{nq} + 4k^2\tau\beta\eta^2\big].
$$

(18)

However, to apply (18), we need an additional result to upper bound the term $\|\tilde{w}^{(t,k-1)} - w^*\|$, summarized as the following lemma.

**Lemma 5.** *With the initialization $\bar{w}^{(0)}$, when $\eta < \min\{\frac{1}{\sqrt{24}\beta K}, \frac{1}{20\beta}, \frac{1}{2\beta + 3K\beta/(nq)}\}$, for any $k \in [0 : K - 1]$,*

$$
\mathbb{E}[\|\tilde{w}^{(t,k)} - w^*\|^2] \leq \|\bar{w}^{(0)} - w^*\|^2 + 8t\beta\eta^3 K^3\tau + (t-1)\big(\bar{Q} + \frac{12K^4\beta^2\eta^4\tau + 3K^2\eta^2\tau}{nq}\big).
$$

From Lemma 5, we also have a global bound that for any $t \in [1 : T]$ and $k \in [0 : K]$,

$$
\mathbb{E}[\|\tilde{w}^{(t,k)} - w^*\|^2] \leq \|\bar{w}^{(0)} - w^*\|^2 + T\big(8\beta\eta^3 K^3\tau + \big(\bar{Q} + \frac{12K^4\beta^2\eta^4\tau + 3K^2\eta^2\tau}{nq}\big)\big). \quad (19)
$$

Now, for any $t_0 \in [1 : T]$ and $k_0 \in [0 : K-1]$, if we select $u = \tilde{w}^{(t_0,k_0)}$, stemmed from (18),

$$\frac{\sum_{(t,k)\in\mathcal{C}} \mathbb{E}[F(\tilde{w}^{(t,k)}) - F(\tilde{w}^{(t_0,k_0)})]}{(T-t_0+1)K - k_0} \leq 3\eta\tau/(nq) + 4K^2\beta\tau\eta^2$$
$$+ \frac{(T-t_0+1)\bar{\mathcal{Q}}}{2\eta((T-t_0+1)K - k_0)} + \frac{5\eta\beta^2 \sum_{(t,k)\in\mathcal{C}} \mathbb{E}[\|\tilde{w}^{(t,k)} - w^*\|^2]}{(T-t_0+1)K - k_0}, \tag{20}$$

where $\mathcal{C} = \big((t_0,k), k = k_0, \cdots, K-1\big) \cup \big((t,k), t = t_0+1, \cdots, T, k = 0, \cdots, K-1\big)$. Finally, as we are concerning about the utility of $\mathcal{F}(\bar{w}^{(T)})$, we need to virtually implement one more gradient descent step on $\bar{w}^{(T)}$ to get an upper bound of $F(\bar{w}^{(T)}) - F(w^*)$. To be specific, we imagine one additional full gradient descent using the entire set on $\bar{w}^{(T)}$, and for any $u$, we have that

$$\|\tilde{w}^{(T+1,1)} - u\|^2 = \|\bar{w}^{(T)} - u - \eta \cdot \frac{\sum_{i=1}^{n} \nabla f_i(\bar{w}^{(T)})}{n}\|^2$$
$$\leq \|\bar{w}^{(T)} - u\|^2 - 2\eta\big(F(\bar{w}^{(T)}) - F(u)\big) + \eta^2\|\nabla F(\bar{w}^{(T)}) - \nabla F(w^*)\|^2$$
$$\leq \|\bar{w}^{(T)} - u\|^2 - 2\eta\big(F(\bar{w}^{(T)}) - F(u)\big) + \min \eta^2\{\beta^2\|\bar{w}^{(T)} - w^*\|^2, 2\beta(F(\bar{w}^{(T)}) - F(w^*))\}. \tag{21}$$

Therefore, let $u = w^*$ and we can combine (16) and (21) to produce the following. Since we assume $(2\eta - 20\eta^2\beta) \geq \eta$ which also implies $2(\eta - \eta^2\beta) \geq \eta$, we have

$$\mathbb{E}\big[\frac{\sum_{t=1}^{T}\sum_{k=1}^{K} \big(F(\tilde{w}^{(t,k-1)}) - F(w^*)\big) + \big(F(\bar{w}^{(T)}) - F(w^*)\big)}{TK+1}\big]$$
$$\leq \frac{\|\bar{w}^{(0)} - w^*\|^2}{\eta(TK+1)} + \big(\frac{6\eta\tau}{nq} + 8\beta K^2\tau\eta^2\big) + \bar{\mathcal{Q}}/(\eta K). \tag{22}$$

Similarly, for (20), it is noted that conditional on $\bar{w}^{(t-1)}$, we have that

$$\mathbb{E}[\|\hat{w}^{(t,k)} - u\|^2] = \mathbb{E}[\|\hat{w}^{(t,k)} - \tilde{w}^{(t,k)}\|^2] + \|\tilde{w}^{(t,k)} - u\|^2, \tag{23}$$

and for $\mathbb{E}[\|\hat{w}^{(t,k)} - \tilde{w}^{(t,k)}\|^2]$ for any $t$ and $k$,

$$\mathbb{E}[\|\hat{w}^{(t,k)} - \tilde{w}^{(t,k)}\|^2] = \mathbb{E}[\|(\hat{w}^{(t,k)} - \bar{w}^{(t-1)}) - (\tilde{w}^{(t,k)} - \bar{w}^{(t-1)})\|^2]$$
$$= \eta^2 \mathbb{E}[\|\sum_{i=1}^{n} \frac{(\mathbf{1}^{(t)} - q)}{nq} \cdot \sum_{l=0}^{k-1} \nabla f_i(w_i^{(t,k)})\|^2] \leq \frac{\eta^2 k(q-q^2)}{n^2 q^2} \cdot \sum_{i=1}^{n}\sum_{l=0}^{k-1} \|\nabla f_i(w_i^{(t,l)})\|^2$$
$$= \frac{\eta^2 k(q-q^2)}{n^2 q^2} \cdot \sum_{i=1}^{n}\sum_{l=0}^{k-1} \|\nabla f_i(w_i^{(t,l)}) - \nabla f_i(\tilde{w}^{(t,l)}) + \nabla f_i(\tilde{w}^{(t,l)}) - F(\tilde{w}^{(t,l)}) + \nabla F(\tilde{w}^{(t,l)}) - \nabla F(w^*)\|^2)$$
$$\leq \frac{3k\eta^2}{n^2 q} \cdot \sum_{i=1}^{n}\sum_{l=0}^{k-1} \big(\beta^2\|w_i^{(t,l)} - \tilde{w}^{(t,l)}\|^2 + \beta^2\|\tilde{w}^{(t,l)} - w^*\|^2 + \tau\big)$$
$$\leq \frac{3K\eta^2}{nq}\big(4\beta^2 K^3\tau\eta^2 + K\tau + \sum_{l=0}^{k-1} \beta^2\|\tilde{w}^{(t,l)} - w^*\|^2\big). \tag{24}$$

where the last line of (24) we apply Lemma 5. Therefore, by replacing $\mathbb{E}[\|\hat{w}^{(t,k)} - u\|^2]$ with $\mathbb{E}[\|\hat{w}^{(t,k)} - \tilde{w}^{(t,k)}\|^2] + \|\tilde{w}^{(t,k)} - u\|^2$ in (18), we have that

$$\mathbb{E}[F(\tilde{w}^{(t,k-1)}) - F(u)] \leq \mathbb{E}\big[\frac{\|\tilde{w}^{(t,k-1)} - u\|^2 - \|\tilde{w}^{(t,k)} - u\|^2 + \|\hat{w}^{(t,k-1)} - \tilde{w}^{(t,k-1)}\|^2 - \|\hat{w}^{(t,k)} - \tilde{w}^{(t,k-1)}\|^2}{2\eta}$$
$$+ 5\eta\beta^2\|\tilde{w}^{(t,k-1)} - w^*\|^2 + \frac{3\eta\tau}{nq} + 4K^2\beta\tau\eta^2\big]. \tag{25}$$

Now, we let $u = \tilde{w}^{(t_0,k_0)}$ in (21) and (25), combining (24) we have

$$\frac{\sum_{t=t_0}^{T}\sum_{k=k_0}^{K-1}\mathbb{E}[F(\tilde{w}^{(t,k)}) - F(\tilde{w}^{(t_0,k_0)})] + \mathbb{E}[F(\bar{w}^{(T)}) - F(\tilde{w}^{(t_0,k_0)})]}{((T-t_0+1)K - k_0 + 1)}$$

$$\leq 3\eta\tau/(nq) + 4K^2\beta\tau\eta^2 + \frac{(T-t_0+1)\bar{\mathcal{Q}}}{2\eta((T-t_0+1)K - k_0 + 1)}$$

$$+ \frac{\frac{3K\eta}{nq}\left(4\beta^2 K^3\tau\eta^2 + K\tau + \sum_{l=0}^{k-1}\beta^2\|\tilde{w}^{(t,l)} - w^*\|^2\right)}{2((T-t_0+1)K - k_0 + 1)}$$

$$+ \frac{5\eta\beta^2\left(\sum_{t=t_0}^{T}\sum_{k=k_0+1}^{K}\mathbb{E}[\|\tilde{w}^{(t,k)} - w^*\|^2] + \mathbb{E}[\|\bar{w}^{(T)} - w^*\|^2]\right)}{(T-t_0+1)K - k_0 + 1}. \tag{26}$$

Now, we can apply the last-iterate convergence rate trick.

**Lemma 6.** *For any sequence $y_i$, $i = 1, 2, \cdots, M$,*

$$y_M = \frac{\sum_{j=1}^{M} y_j}{M} + \sum_{j=1}^{M-1}\frac{\sum_{l=M-j+1}^{M}(y_l - y_{M-j})}{j(j+1)} \tag{27}$$

One can easily verify the identity in Lemma 6.

If we take $y_j = \mathbb{E}[F(\tilde{w}^{(t,k)}) - F(w^*)]$ and $z_j = \mathbb{E}[\|\tilde{w}^{(t,k)} - w^*\|^2]$, for $j = (t-1)K + k$ and let $M = TK + 1$ where $y_{TK+1} = \mathbb{E}[F(\bar{w}^{(T)}) - F(w^*)]$ and $z_{TK+1} = \mathbb{E}[\|\bar{w}^{(T)} - w^*\|^2]$, combined with (22),(26) and Lemma 6, we have that

$$y_{TK+1} = \mathbb{E}[F(\bar{w}^{(T)}) - F(w^*)] \tag{28}$$

$$= \frac{\sum_{j=1}^{TK} y_j}{TK + 1} + \sum_{j=1}^{TK}\frac{1}{j+1}\cdot\frac{\sum_{l=TK+2-j}^{TK+1}(y_l - y_{TK+1-j})}{j} \tag{29}$$

$$\leq \left\{\frac{\|\bar{w}^{(0)} - w^*\|^2}{\eta(TK+1)} + \left(\frac{6\eta\tau}{nq} + 8\beta K^2\tau\eta^2\right) + \bar{\mathcal{Q}}/(\eta K)\right\} \tag{30}$$

$$+ \sum_{j=1}^{TK}\left\{\frac{1}{j+1}\cdot\left(\frac{3\eta\tau}{nq} + 4\beta K^2\tau\eta^2 + \frac{\bar{\mathcal{Q}}}{2\eta} + \frac{12K^4\eta^3\beta^2\tau}{2nq} + \frac{3K^2\eta\tau}{2nq} + \frac{3K^2\eta}{nq}\max_l\{z_l\}\right) + 5\eta\beta^2\frac{\sum_{l=TK-j+2}^{TK+1} z_l}{j(j+1)}\right\} \tag{31}$$

$$\leq \frac{\|\bar{w}^{(0)} - w^*\|^2}{\eta(TK+1)} + \log(TK+1)\left(\frac{6\eta\tau}{nq} + 8\beta K^2\tau\eta^2 + \bar{\mathcal{Q}}/\eta + \frac{12K^4\eta^3\beta^2\tau}{2nq} + \frac{3K^2\eta\tau}{2nq} + \frac{3K^2\eta}{nq}\max_l\{z_l\}\right) \tag{32}$$

$$+ (5\eta\beta^2)\sum_{j=1}^{TK}\left(\frac{1}{j} - \frac{1}{TK+1}\right)\cdot z_{TK-j+2} \tag{33}$$

In (30), we apply (22) on $\frac{\sum_{j=1}^{TK} y_j}{TK+1}$. In (31), we apply the results in (26) and $\frac{(T-t_0+1)\bar{\mathcal{Q}}}{2\eta((T-t_0+1)K - k_0+1)} \leq \frac{\bar{\mathcal{Q}}}{2\eta}$, since the number of iterates is always no less than the number of synchronization in any time interval. In (33), we use the fact that $\sum_{j=1}^{TK}\frac{1}{j+1} \leq \log(TK+1)$ and as assumed $\log(TK) \geq 2$. Now, with the assumption that $K^2 = O(nq)$, (33) can be further bounded as

$$y_{TK+1} < O(1)\cdot\left(\frac{\|\bar{w}^{(0)} - w^*\|^2}{\eta(TK+1)} + \log(TK+1)\left(\frac{\eta\tau}{nq} + K^2\tau\eta^2 + \bar{\mathcal{Q}}/\eta + \tau\eta\right) + \eta\left(\sum_{j=1}^{TK}\frac{1}{j}\right)\cdot\max_l\{z_l\}\right) \tag{34}$$

$$\leq O(1)\cdot\left(\frac{\|\bar{w}^{(0)} - w^*\|^2}{\eta(TK+1)} + \log(TK+1)\left(\frac{\eta\tau}{nq} + K^2\tau\eta^2 + \bar{\mathcal{Q}}/\eta + \tau\eta\right)\right) \tag{35}$$

$$+ \eta(\log(TK)+1)\left(\|\bar{w}^{(0)} - w^*\|^2 + T\left(\beta\eta^3 K^3\tau + \frac{K^4\beta^2\eta^4\tau + K^2\eta^2\tau}{nq} + \bar{\mathcal{Q}}\right)\right). \tag{36}$$

In (36), we apply Lemma 5 and (19). Thus, we complete the proof.

## B.2 PROOF OF LEMMA 2

Conditional on $\bar{w}^{(t-1)}$, we have that

$$\mathbb{E}[\|\frac{\sum_{i=1}^n \eta \mathbf{1}_i^{(t)} \nabla f_i(w_i^{(t,k-1)})}{nq}\|^2]$$

$$= \mathbb{E}[\|\frac{\sum_{i=1}^n \eta \mathbf{1}_i^{(t)} \nabla f_i(w_i^{(t,k-1)})}{nq} - \frac{\sum_{i=1}^n \eta \nabla f_i(w_i^{(t,k-1)})}{n} + \frac{\sum_{i=1}^n \eta \nabla f_i(w_i^{(t,k-1)})}{n}\|^2]$$

$$\leq 2 \cdot \mathbb{E}[\|\frac{\sum_{i=1}^n \eta (\mathbf{1}_i^{(t)} - q) \nabla f_i(w_i^{(t,k-1)})}{nq}\|^2] + 2 \cdot \|\frac{\sum_{i=1}^n \eta \nabla f_i(w_i^{(t,k-1)})}{n}\|^2 \quad (37)$$

$$= \frac{2(q - q^2) \sum_{i=1}^n \|\eta \nabla f_i(w_i^{(t,k-1)})\|^2}{(nq)^2} + 2 \cdot \|\frac{\sum_{i=1}^n \eta \nabla f_i(w_i^{(t,k-1)})}{n}\|^2$$

$$\leq \frac{2\eta^2 \sum_{i=1}^n \|\nabla f_i(w_i^{(t,k-1)})\|^2}{n^2 q} + 2\eta^2 \|\frac{\sum_{i=1}^n \nabla f_i(w_i^{(t,k-1)})}{n}\|^2.$$

In the fourth line of (37), we use the fact that $\mathbf{1}_{[1:n]}^{(t)}$ are i.i.d. Bernoulli variable of mean $q$, and thus $\mathbb{E}[(\mathbf{1}_i^{(t)} - q)^2] = q(1-q)$ and $\mathbb{E}[(\mathbf{1}_i^{(t)} - q) \cdot (\mathbf{1}_j^{(t)} - q)] = 0$ for $i \neq j$. As for $\sum_{i=1}^n \|\nabla f_i(w_i^{(t,k-1)})\|^2$, we can further bound it as follows,

$$\sum_{i=1}^n \|\nabla f_i(w_i^{(t,k-1)}) - \nabla f_i(\tilde{w}^{(t,k-1)}) + \nabla f_i(\tilde{w}^{(t,k-1)}) - \nabla f_i(w^*) + \nabla f_i(w^*)\|^2 \quad (38)$$

$$\leq 3 \sum_{i=1}^n \left(\beta^2 \|w_i^{(t,k-1)} - \tilde{w}^{(t,k-1)}\|^2 + 2\beta \mathcal{D}_{f_i}(\tilde{w}^{(t,k-1)}, w^*) + \|\nabla f_i(w^*)\|^2\right) \quad (39)$$

$$\leq 3\beta^2 \sum_{i=1}^n \|w_i^{(t,k-1)} - \tilde{w}^{(t,k-1)}\|^2 + 6\beta n \left(F(\tilde{w}^{(t,k-1)}) - F(w^*)\right) + 3n\tau. \quad (40)$$

In (39), we apply AM-GM inequality again and use the property that for convex and $\beta$-smooth function $f_i(w)$, it holds that $\|\nabla f_i(x) - \nabla f_i(y)\|^2 \leq 2\beta \mathcal{D}_{f_i}(x, y)$, where $\mathcal{D}_{f_i}(x, y) = f(x) - f(y) - \langle \nabla f(y), x - y \rangle$ is the Bregman divergence. In (40), we use the fact that $\nabla F(w^*) = 0$ and due to Assumption 1, the variance $\sum_{i=1}^n \|\nabla f_i(w^*) - \nabla F(w^*)\|^2 = \sum_{i=1}^n \|\nabla f_i(w^*)\|^2 \leq n\tau$. When we apply similar decomposition tricks in (40) to the term $\|\frac{\sum_{i=1}^n \nabla f_i(w_i^{(t,k-1)})}{n}\|^2$,

$$\|\frac{\sum_{i=1}^n \nabla f_i(w_i^{(t,k-1)})}{n}\|^2$$

$$\leq \|\frac{\sum_{i=1}^n \nabla f_i(w_i^{(t,k-1)}) - \nabla f_i(\tilde{w}^{(t,k-1)}) + \nabla f_i(\tilde{w}^{(t,k-1)}) - \nabla f_i(w^*) + \nabla f_i(w^*)}{n}\|^2$$

$$\leq 2\left(\|\frac{\sum_{i=1}^n \nabla f_i(w_i^{(t,k-1)}) - \nabla f_i(\tilde{w}^{(t,k-1)})}{n}\|^2 + \|\frac{\sum_{i=1}^n \nabla f_i(\tilde{w}^{(t,k-1)}) - \nabla f_i(w^*)}{n}\|^2\right)$$

$$\leq \frac{2\beta^2 \sum_{i=1}^n \|w_i^{(t,k-1)} - \tilde{w}^{(t,k-1)}\|^2}{n} + 4\beta\left(F(\tilde{w}^{(t,k-1)}) - F(w^*)\right),$$

since $\nabla F(w^*) = \frac{1}{n} \cdot \sum_{i=1}^n \nabla f_i(w^*) = 0$. Thus, (37) can be further bounded as follows:

$$\mathbb{E}[\|\frac{\sum_{i=1}^n \eta \mathbf{1}_i^{(t)} \nabla f_i(w_i^{(t,k-1)})}{nq}\|^2]$$

$$\leq \frac{10\eta^2 \beta^2}{n} \sum_{i=1}^n \|w_i^{(t,k-1)} - \tilde{w}^{(t,k-1)}\|^2 + 20\beta\eta^2 (F(\tilde{w}^{(t,k-1)}) - F(w^*)) + \frac{6\eta^2 \tau}{nq}. \quad (41)$$

Here, we use the fact that $q \geq 1/n$ and thus $\frac{1}{n^2 q} \leq \frac{1}{n}$. Meanwhile, it is noted that $\|\nabla f_i(\tilde{w}^{(t,k-1)}) - \nabla f_i(w^*)\|^2$ can also be bounded by $\beta^2 \|\tilde{w}^{(t,k-1)} - w^*\|^2$ alternatively due to the smooth assumption. Thus, by replacing $2\beta(F(\tilde{w}^{(t,k-1)}) - F(w^*))$ in (39) and (41) with $\beta^2 \|\tilde{w}^{(t,k-1)} - w^*\|^2$, we complete the proof.

### B.3 PROOF OF LEMMA 3

Based on the Poisson sampling assumption, conditional on $\bar{w}^{(t-1)}$,

$$\mathbb{E}\Big[-\frac{2}{nq}\cdot\sum_{i=1}^{n}\eta\mathbf{1}_i^{(t)}\langle\tilde{w}^{(t,k-1)}-u,\nabla f_i(w_i^{(t,k-1)})\rangle\Big]=-\frac{2\eta}{n}\Big[\sum_{i=1}^{n}\langle\tilde{w}^{(t,k-1)}-u,\nabla f_i(w_i^{(t,k-1)})\rangle\Big].$$

For each $i$, it is noted that

$$\begin{aligned}
&-\langle\tilde{w}^{(t,k-1)}-u,\nabla f_i(w_i^{(t,k-1)})\rangle\\
&=-\langle w_i^{(t,k-1)}-u,\nabla f_i(w_i^{(t,k-1)})\rangle-\langle\tilde{w}^{(t,k-1)}-w_i^{(t,k-1)},\nabla f_i(w_i^{(t,k-1)})\rangle\\
&\leq f_i(u)-f_i(w_i^{(t,k-1)})+f_i(w_i^{(t,k-1)})-f_i(\tilde{w}^{(t,k-1)})+\frac{\beta}{2}\|w_i^{(t,k-1)}-\tilde{w}^{(t,k-1)}\|^2.
\end{aligned} \tag{42}$$

In (42), we use the following facts. First, for smooth and convex function $f_i$, $\mathcal{D}_{f_i}(u,w_i^{(t,k-1)})\geq0$ and thus $-\langle w_i^{(t,k-1)}-u,\nabla f_i(w_i^{(t,k-1)})\rangle\leq f_i(u)-f_i(w_i^{(t,k-1)})$. Second, for the term $-\langle\tilde{w}^{(t,k-1)}-w_i^{(t,k-1)},\nabla f_i(w_i^{(t,k-1)})\rangle$, we use the classic smooth inequality where

$$f_i(\tilde{w}^{(t,k-1)})\leq f_i(w_i^{(t,k-1)})+\langle\tilde{w}^{(t,k-1)}-w_i^{(t,k-1)},\nabla f_i(w_i^{(t,k-1)})\rangle+\frac{\beta}{2}\|w_i^{(t,k-1)}-\tilde{w}^{(t,k-1)}\|^2.$$

Therefore, by (42), we have that

$$-\frac{2\eta}{n}\Big[\sum_{i=1}^{n}\langle\tilde{w}^{(t,k-1)}-u,\nabla f_i(w_i^{(t,k-1)})\rangle\Big]\leq2\eta\big(F(u)-F(\tilde{w}^{(t,k-1)})\big)+\frac{\beta}{2n}\sum_{i=1}^{n}\|w^{(t,k-1)}-\tilde{w}^{(t,k-1)}\|^2\big).$$

### B.4 PROOF OF LEMMA 4

Given $\bar{w}^{(t-1)}$,

$$\sum_{i=1}^{n}\big[\|w_i^{(t,k)}-\tilde{w}^{(t,k)}\|^2\big]=\eta^2\sum_{i=1}^{n}\Big[\|\sum_{l=0}^{k-1}\nabla f_i(w_i^{(t,l)})-\frac{\sum_{j=1}^{n}\sum_{l=0}^{k-1}\nabla f_j(w_j^{(t,l)})}{n}\|^2\Big] \tag{43}$$

$$\leq3k\eta^2\Big[\sum_{i=1}^{n}\sum_{l=0}^{k-1}\big(\|\nabla f_i(w_i^{(t,l)})-\nabla f_i(\tilde{w}^{(t,l)})\|^2+\|\nabla f_i(\tilde{w}^{(t,l)})-\nabla F(\tilde{w}^{(t,l)})\|^2 \tag{44}$$

$$+\|\nabla F(\tilde{w}^{(t,l)})-\frac{\sum_{j=1}^{n}\nabla f_j(w_j^{(t,l)})}{n}\|^2\big)\Big] \tag{45}$$

$$\leq3k\eta^2\Big[\big(\sum_{i=1}^{n}\sum_{l=0}^{k-1}\beta^2\|w_i^{(t,l)}-\tilde{w}^{(t,l)}\|^2\big)+kn\tau+\sum_{i=1}^{n}\sum_{l=0}^{k-1}\sum_{j=1}^{n}\frac{\beta^2\|\tilde{w}^{(t,l)}-w_j^{(t,l)}\|^2}{n}\Big] \tag{46}$$

$$\leq6k\beta^2\eta^2\sum_{i=1}^{n}\sum_{l=0}^{k-1}\big[\|w_i^{(t,l)}-\tilde{w}^{(t,l)}\|^2\big]+3k^2n\tau\eta^2. \tag{47}$$

In (46), we use Assumption 1 that the variance of stochastic gradient is bounded by $\tau$ and apply the form $\nabla F(\tilde{w}^{(t,l)})=\frac{\sum_{i=1}^{n}\nabla f_i(\tilde{w}^{(t,l)})}{n}$.

Let $M^{(k)}=\mathbb{E}[\sum_{i=1}^{n}\|w_i^{(t,k)}-\tilde{w}^{(t,k)}\|^2]$. Then, from (47), when $n\geq1$, we have an inequality in a form

$$M^{(k)}\leq\eta^2\big(6k\beta^2\sum_{l=0}^{k-1}M^{(l)}+3k^2n\tau\big),$$

where $M^{(0)}=\|\bar{w}^{(t-1)}-\bar{w}^{(t-1)}\|^2=0$. It is not hard to verify that by induction, once $\eta^2<\frac{1}{24\beta^2K^2}$, $M^{(k)}\leq4\eta^2k^2n\tau$.

## B.5 PROOF OF LEMMA 5

To provide more intuition, we start from the case when $t = 1$, $\tilde{w}^{(t,0)} = \bar{w}^{(0)}$ and thus

$$\|\tilde{w}^{(1,k)} - w^*\|^2 = \|\tilde{w}^{(1,k-1)} - w^*\|^2 - 2\eta\langle \frac{\sum_{i=1}^n \nabla f_i(w_i^{(1,k-1)})}{n}, \tilde{w}^{(1,k-1)} - w^*\rangle + \eta^2 \|\frac{\sum_{i=1}^n \nabla f_i(w_i^{(1,k-1)})}{n}\|^2.$$

As a straightforward corollary of Lemma 2, 3 and 4, we can obtain a similar upper bound in a form once $\eta < \min\{\frac{\beta}{\sqrt{24K}}, \frac{1}{2\beta}\}$

$$\|\tilde{w}^{(1,k)} - w^*\|^2 \leq \|\tilde{w}^{(1,k-1)} - w^*\|^2 + 2\eta\big(F(w^*) - F(\tilde{w}^{(t,k-1)}) + \frac{\beta}{2n}\sum_{i=1}^n \|w^{(t,k-1)} - \tilde{w}^{(t,k-1)}\|^2\big)$$

$$+ 2\eta^2\big(\frac{\beta^2 \sum_{i=1}^n \|w_i^{(t,k-1)} - \tilde{w}^{(t,k-1)}\|^2}{n} + 2\beta F(\tilde{w}^{(t,k-1)}) - F(w^*)\big)$$

$$\leq \|\tilde{w}^{(1,k-1)} - w^*\|^2 + 2(\eta - 2\beta\eta^2)(F(w^*) - F(\tilde{w}^{(t,k-1)})) + (\beta\eta + 2\beta^2\eta^2) \cdot 4\eta^2 K^2 \tau$$

$$\leq \|\tilde{w}^{(1,k-1)} - w^*\|^2 + 2(\eta - 2\beta\eta^2)(F(w^*) - F(\tilde{w}^{(t,k-1)})) + 8\beta\eta^3 K^2 \tau.$$

$$(48)$$

In (48), we apply Lemma 4 and use the fact that $\beta\eta + 2\beta^2\eta^2 \leq 2\beta\eta$.

On the other hand, during the synchronization, it is noted that

$$\mathbb{E}[\bar{w}^{(1)}] = \mathbb{E}[\tilde{w}^{(1,K)} + Q^{(1)}] = \mathbb{E}[\tilde{w}^{(1,K)}].$$

Therefore,

$$\mathbb{E}[\|\bar{w}^{(1)} - w^*\|^2] = \mathbb{E}[\|\bar{w}^{(1)} - \tilde{w}^{(1,K)}\|^2] + \|\tilde{w}^{(1,K)} - w^*\|^2.$$

Moreover,

$$\mathbb{E}[\|\bar{w}^{(1)} - \tilde{w}^{(1,K)}\|^2]$$

$$= \mathbb{E}[\eta^2 \|\frac{\sum_{k=1}^K \sum_{i=1}^n (1_i^{(1)} - q)\nabla f_i(w_i^{(1,k-1)})}{nq} - Q^{(1)}\|^2]$$

$$\leq \frac{K\eta^2 \sum_{k=1}^K \sum_{i=1}^n \|\nabla f_i(w_i^{(1,k-1)})\|^2}{n^2 q} + \bar{Q}$$

$$\leq \frac{3K\eta^2 \sum_{k=1}^K \big\{ \sum_{i=1}^n \big(\beta^2 \|w_i^{(1,k-1)} - \tilde{w}^{(1,k-1)}\|^2\big) + 2\beta n(F(\tilde{w}^{(1,k-1)}) - F(w^*)) + n\tau\big\}}{n^2 q} + \bar{Q}$$

$$\leq \frac{3K\eta^2\big(4\beta^2\eta^2 K^3 n\tau + 2\beta n \sum_{k=1}^K (F(\tilde{w}^{(1,k-1)}) - F(w^*)) + Kn\tau\big\}}{n^2 q} + \bar{Q}$$

$$= \frac{12K^4\beta^2\eta^4\tau + 6K\beta\eta^2 \sum_{k=1}^K (F(\tilde{w}^{(1,k-1)}) - F(w^*)) + 3K^2\eta^2\tau}{nq} + \bar{Q}.$$

$$(49)$$

In the fifth line of (49), we apply Lemma 4. From (48),

$$\|\tilde{w}^{(1,K)} - w^*\|^2 \leq \|\bar{w}^{(0)} - w^*\|^2 + 2(\eta - 2\beta\eta^2)\sum_{k=1}^K (F(w^*) - F(\tilde{w}^{(t,k-1)})) + 8\beta\eta^3 K^3 \tau. \quad (50)$$

Now, we combine (49) and (50). Once $2(\eta - 2\beta\eta^2) - \frac{6K\beta\eta^2}{nq} \geq 0$, which implies that $\eta \leq \frac{1}{2\beta + 3K\beta/(nq)}$,

$$\mathbb{E}[\|\bar{w}^{(1)} - w^*\|^2] \leq \|\bar{w}^{(0)} - w^*\|^2 + \frac{12K^4\beta^2\eta^4\tau + 3K^2\eta^2\tau}{nq} + 8\beta\eta^3 K^3 \tau + \bar{Q}.$$

The remainder of the proof for the $\|\tilde{w}^{(t,k)} - w^*\|$ is straightforward as for arbitrary $t$, $\|\tilde{w}^{(t,0)} - w^*\| = \|\bar{w}^{(t-1)} - w^*\|$. Therefore, by induction reasoning, we have the bound claimed.

## C  PROOF OF THEOREM 4: SYNCHRONIZED-ONLY CONVERGENCE OF NOISY LSGD IN NON-CONVEX OPTIMIZATION

Based on the smooth assumption of $F(w)$, we have the following classic inequality,

$$F(\bar{w}^{(t)}) \le F(\bar{w}^{(t-1)}) + \langle \nabla F(\bar{w}^{(t-1)}), \bar{w}^{(t)} - \bar{w}^{(t-1)} \rangle + \frac{\beta}{2} \|\bar{w}^{(t)} - \bar{w}^{(t-1)}\|^2$$

$$= F(\bar{w}^{(t-1)}) - \langle \nabla F(\bar{w}^{(t-1)}), \frac{\eta}{nq} \sum_{i \in S^{(t)}} \sum_{k=0}^{K-1} \nabla f_i(w_i^{(t,k)}) - Q^{(t)} \rangle$$

$$+ \frac{\beta}{2} \| \frac{\eta}{nq} \sum_{i \in S^{(t)}} \sum_{k=0}^{K-1} \nabla f_i(w_i^{(t,k)}) - Q^{(t)} \|^2$$

$$= F(\bar{w}^{(t-1)})$$

$$- \frac{\eta}{2} \Big( \sum_{k=0}^{K-1} \big( \|\nabla F(\bar{w}^{(t-1)})\|^2 + \| \frac{1}{nq} \sum_{i \in S^{(t)}} \nabla f_i(w_i^{(t,k)})\|^2 - \|\nabla F(\bar{w}^{(t-1)}) - \frac{1}{nq} \sum_{i \in S^{(t)}} \nabla f_i(w_i^{(t,k)})\|^2 \big) \Big)$$

$$+ \langle \nabla F(\bar{w}^{(t-1)}), Q^{(t)} \rangle + \frac{\beta}{2} \| \frac{\eta}{nq} \sum_{i \in S^{(t)}} \sum_{k=0}^{K-1} \nabla f_i(w_i^{(t,k)}) - Q^{(t)} \|^2. \tag{51}$$

In (51), we simply use the fact that $\langle a, b \rangle = \frac{\|a\|^2 + \|b\|^2 - \|a-b\|^2}{2}$. For notation simplicity, we will use $g_i^{(t,k)} = \nabla f_i(w_i^{(t,k)})$ and $g^{(t,k)} = \frac{1}{nq} \cdot \sum_{i \in S_t} \nabla f_i(w_i^{(t,k)}) = \frac{1}{nq} \cdot \sum_{i \in S_t} g_i^{(t,k)}$ in the following. Using the generalized AM-GM inequality, where $\langle a, b \rangle \le \frac{1}{2} \big( \gamma \|a\|^2 + \frac{1}{\gamma} \|b\|^2 \big)$ for any $\gamma > 0$, on $\langle \nabla F(w^{(t-1)}), Q^{(t)} \rangle$, we have that

$$\langle \nabla F(w^{(t-1)}), Q^{(t)} \rangle \le \frac{\eta}{4} \|\nabla F(w^{(t-1)})\|^2 + \frac{1}{\eta} \|Q^{(t)}\|^2. \tag{52}$$

Similarly,

$$\frac{\beta}{2} \| \frac{\eta}{nq} \sum_{i \in S_t} \sum_{k=0}^{K-1} g_i^{(t,k)} - Q^{(t)} \|^2 \le \beta \big( \eta^2 \| \frac{1}{nq} \sum_{i \in S_t} \sum_{k=0}^{K-1} g_i^{(t,k)} \|^2 + \|Q^{(t)}\|^2 \big). \tag{53}$$

Thus, putting together, we have the following by rearranging the terms in (51),

$$(\frac{\eta K}{2} - \frac{\eta}{4}) \|\nabla F(\bar{w}^{(t-1)})\|^2 \le F(\bar{w}^{(t-1)}) - F(\bar{w}^{(t)}) - \underbrace{\big( \frac{\eta}{2} \sum_{k=0}^{K-1} \|g^{(t,k)}\|^2 - \beta \eta^2 \| \sum_{k=0}^{K-1} g^{(t,k)} \|^2 \big)}_{(A)}$$

$$+ \frac{\eta}{2} \sum_{k=0}^{K-1} \|\nabla F(\bar{w}^{(t-1)}) - g^{(t,k)}\|^2 + (\frac{1}{\eta} + \beta) \|Q^{(t)}\|^2. \tag{54}$$

Still by AM-GM inequality, it is noted that $\| \sum_{k=0}^{K-1} g^{(t,k)} \|^2 \le K \sum_{k=0}^{K-1} \|g^{(t,k)}\|^2$ and therefore term (A) is lower bounded by $(\frac{\eta}{2} - \beta \eta^2 K) \sum_{k=0}^{K-1} \|g^{(t,k)}\|^2$. For a sufficiently small learning rate $\eta$, term (A) is non-negative. Thus, to upper bound $\|\nabla F(w^{(t)})\|^2$, it suffices to keep track of $\|\nabla F(w^{(t)}) - g^{(t,k)}\|^2$.

Now, we imagine the scenario that each agent participates in the $t$-th phase without Poisson sampling and each produces intermediate $w_i^{(t,k)}$ for $i = 1, 2, \cdots, n$ and $k = 1, 2, \cdots, K$. Let $\tilde{w}^{(t,k)} = \frac{1}{n} \sum_{i=1}^n w_i^{(t,k)}$. It is not hard to observe that conditional on $\bar{w}^{(t-1)}$, $\mathbb{E}[\tilde{w}^{(t,k)} - \bar{w}^{(t-1)}] =$

$-\eta\mathbb{E}[\sum_{l=0}^{k-1} g^{(t,l)}]$. On the other hand, by AM-GM inequality again,

$$
\begin{aligned}
&\|\nabla F(w^{(t-1)}) - g^{(t,k)}\|^2 \\
&\leq 2\big(\|\nabla F(\bar{w}^{(t-1)}) - \nabla F(\tilde{w}^{(t,k)})\|^2 + \|\nabla F(\tilde{w}^{(t,k)}) - g^{(t,k)}\|^2\big) \\
&\leq 2\big(\beta^2\|\bar{w}^{(t-1)} - \tilde{w}^{(t,k)}\|^2 + \|\nabla F(\tilde{w}^{(t,k)}) - g^{(t,k)}\|^2\big) \\
&= 2\big(\beta^2\|\bar{w}^{(t-1)} - \tilde{w}^{(t,k)}\|^2 + \|\frac{\sum_{i=1}^{n}(q - 1_i^{(t)})\big(\nabla f_i(\tilde{w}^{(t,k)}) - \nabla f_i(w_i^{(t,k)})\big)}{nq}\|^2\big).
\end{aligned}
\tag{55}
$$

In (55), we use the $\beta$-smooth assumption on $\nabla F(w)$, and $1_i^{(t)}$ is an indicator which equals 1 iff the $i$-th worker/agent is selected in the $t$-th phase with probability $q$, otherwise 0. We first handle the first term $\beta^2\|\bar{w}^{(t)} - \tilde{w}^{(t,k)}\|^2$. With expectation conditional on $\bar{w}^{(t-1)}$,

$$
\begin{aligned}
\mathbb{E}[\|\bar{w}^{(t-1)} - \tilde{w}^{(t,k)}\|^2] &= \mathbb{E}[\eta^2\|\sum_{l=0}^{k-1} g^{(t,l)}\|^2] - \mathbb{E}\big[\| - (\eta\sum_{l=0}^{k-1} g^{(t,l)}) - (\bar{w}^{(t-1)} - \tilde{w}^{(t,k)})\|^2\big] \\
&\leq k\eta^2\sum_{l=0}^{k-1}\mathbb{E}[\|g^{(t,l)}\|^2]
\end{aligned}
\tag{56}
$$

In (56), we use the following fact about the variance and second moment: for a random vector $v$ whose mean is $\mu$, $\mathbb{E}[\|v\|^2] = \mathbb{E}[\|v - \mu\|^2] + \|\mu\|^2$. As mentioned above, the expectation conditional on $\bar{w}^{(t-1)}$ $\mathbb{E}[\tilde{w}^{(t,k)} - \bar{w}^{(t-1)}] = -\eta\mathbb{E}[\sum_{l=0}^{k-1} g^{(t,l)}]$. Therefore,

$$
2\beta^2\sum_{k=1}^{K}\mathbb{E}[\|\bar{w}^{(t-1)} - \tilde{w}^{(t,k)}\|^2] \leq 2\beta^2\sum_{k=1}^{K}k\eta^2\sum_{l=0}^{k-1}\mathbb{E}[\|g^{(t,l)}\|^2] \leq 2\beta^2\eta^2 K^2\sum_{k=0}^{K-1}\mathbb{E}[\|g^{(t,k)}\|^2].
\tag{57}
$$

Now, combined the same term $\mathbb{E}[\|g^{(t,k)}\|^2]$ in (57) with (A), it is not hard to verifiy that, once $\frac{\eta}{2} - \beta\eta^2 K - \beta^2\eta^3 K^2 \geq 0$, which holds when $\eta < \frac{1}{4\beta K}$, then the expectation

$$
\mathbb{E}\big[\frac{\eta}{2} \cdot 2\beta^2 K^2\eta^2\sum_{k=0}^{K-1}\|\sum_{l=0}^{k} g^{(t,l)}\|^2 - (A)\big] \leq 0.
$$

Now, we move our focus to the second term $\|\frac{1}{nq} \cdot \sum_{i=1}^{n}(q - 1_i^{(t)})\big(\nabla f_i(\tilde{w}^{(t,k)}) - \nabla f_i(w_i^{(t,k)})\big)\|^2$ in (55).

Based on the assumption on Poisson sampling, $1_i^{(t)}$ is independent and $\mathbb{E}[1_i^{(t)}] = q$ for $i = 1, 2, \cdots, n$. Morevoer, $\mathbb{E}[(1_i^{(t)} - q)^2] = q - q^2 < q$. Therefore, with expectation,

$$
\begin{aligned}
&\sum_{k=0}^{K-1}\mathbb{E}\big[\|\frac{\sum_{i=1}^{n}(q - 1_i^{(t)})\big(\nabla f_i(\tilde{w}^{(t,k)}) - \nabla f_i(w_i^{(t,k)})\big)}{nq}\|^2\big] \\
&= \sum_{k=0}^{K-1}\sum_{i=1}^{n}\frac{(q - q^2)\mathbb{E}[\|\nabla f_i(\tilde{w}^{(t,k)}) - \nabla f_i(w_i^{(t,k)})\|^2]}{(nq)^2} \leq \sum_{k=0}^{K-1}\sum_{i=1}^{n}\frac{\beta^2\mathbb{E}[\|\tilde{w}^{(t,k)} - w_i^{(t,k)}\|^2]}{n^2 q}.
\end{aligned}
\tag{58}
$$

In (58), we use the fact for $n$ random independent vectors $v_{[1:n]}$ of zero mean, $\mathbb{E}[\|\sum_{i=1}^{n} v_i\|^2] = \sum_{i=1}^{n}\mathbb{E}[\|v_i\|^2]$. On the other hand, we can apply the results of Lemma 4 to upper bound $\sum_{i=1}^{n}\mathbb{E}\big[\|w_i^{(t,k)} - \tilde{w}^{(t,k)}\|^2\big]$ by $4\eta^2 k^2 n\tau$ once $\eta < \min\{\frac{\beta}{\sqrt{24K}}, \frac{1}{20\beta}\}$. Now, back to (58), we have that

$$
\sum_{k=0}^{K-1}\sum_{i=1}^{n}\frac{\beta^2\mathbb{E}[\|\tilde{w}^{(t,k)} - w_i^{(t,k)}\|^2]}{n^2 q} \leq \frac{4\eta^2\tau\beta^2 K^3}{nq}.
$$

With the above preparation, we are finally ready to complete the proof. Back to (54), conditional on $w^{(t-1)}$, with expectation we have that

$$(\frac{\eta K}{2} - \frac{\eta}{4})\|\nabla F(\bar{w}^{(t-1)})\|^2 \leq \mathbb{E}[F(\bar{w}^{(t-1)}) - F(\bar{w}^{(t)})] - (\frac{\eta}{2} - \beta\eta^2 K - \beta^2\eta^3 K^2) \sum_{k=0}^{K-1} \mathbb{E}[\|g^{(t,k)}\|^2]$$

$$+ \frac{\eta}{2} \cdot \frac{8\eta^2\tau\beta^2 K^3}{nq} + (\frac{1}{\eta} + \beta)\|Q^{(t)}\|^2.$$

(59)

Summing up both sides of (59) for $t = 1, 2, ..., T$, with unconditional expectation and averaging, since $\eta K/2 - \eta/4 \geq \eta K/4$ for $K \geq 1$, we obtain that once $\eta < \min\{\frac{\beta}{\sqrt{24K}}, \frac{1}{4\beta K}, \frac{1}{20\beta}\}$,

$$\mathbb{E}[\frac{\sum_{t=1}^{T} \|\nabla F(\bar{w}^{(t-1)})\|^2}{T}] \leq \frac{4F(\bar{w}^{(0)})}{TK\eta} + \frac{16\eta^2\tau\beta^2 K^2}{nq} + \frac{(1 + \beta\eta) \sum_{t=1}^{T} \mathbb{E}[\|Q^{(t)}\|^2]}{\eta^2 KT}.$$

Alternatively, especially when the perturbation $Q^{(t)}$ is independent and of zero-mean, we may consider another bound derived as follows. Still, based on the smooth assumption of $F(w)$, if we focus on each cross term between $\nabla F(\bar{w}^{(t-1)})$ and $\nabla f_i(w_i^{(t,k)})$, we have

$$F(\bar{w}^{(t)}) \leq F(\bar{w}^{(t-1)}) + \langle \nabla F(\bar{w}^{(t-1)}), \bar{w}^{(t)} - \bar{w}^{(t-1)} \rangle + \frac{\beta}{2}\|\bar{w}^{(t)} - \bar{w}^{(t-1)}\|^2$$

$$= F(\bar{w}^{(t-1)}) - \langle \nabla F(\bar{w}^{(t-1)}), \frac{\eta}{nq} \sum_{i \in S^{(t)}} \sum_{k=0}^{K-1} \nabla f_i(w_i^{(t,k)}) - Q^{(t)} \rangle$$

$$+ \frac{\beta}{2}\|\frac{\eta}{nq} \sum_{i \in S^{(t)}} \sum_{k=0}^{K-1} \nabla f_i(w_i^{(t,k)}) - Q^{(t)}\|^2$$

$$= F(\bar{w}^{(t-1)})$$

$$- \frac{\eta}{2nq} \cdot \Big( \sum_{i \in S^{(t)}} \sum_{k=0}^{K-1} \big(\|\nabla F(\bar{w}^{(t-1)})\|^2 + \|\nabla f_i(w_i^{(t,k)})\|^2 - \|\nabla F(\bar{w}^{(t-1)}) - \nabla f_i(w_i^{(t,k)})\|^2\big)\Big)$$

$$+ \langle \nabla F(\bar{w}^{(t-1)}), Q^{(t)} \rangle + \frac{\beta}{2}\|\frac{\eta}{nq} \sum_{i \in S^{(t)}} \sum_{k=0}^{K-1} \nabla f_i(w_i^{(t,k)}) - Q^{(t)}\|^2.$$

(60)

With a similar reasoning as (53), we have the following by rearranging the terms in (60),

$$\frac{\eta K B_t}{2nq}\|\nabla F(\bar{w}^{(t-1)})\|^2 \leq F(\bar{w}^{(t-1)}) - F(\bar{w}^{(t)}) - \underbrace{(\frac{\eta}{2nq} - \frac{\beta\eta^2 B_t K}{(nq)^2}) \sum_{i \in S^{(t)}} \sum_{k=0}^{K-1} \|g_i^{(t,k)}\|^2}_{(A)}$$

$$+ \frac{\eta}{2nq} \sum_{i \in S^{(t)}} \sum_{k=0}^{K-1} \|\nabla F(\bar{w}^{(t-1)}) - g_i^{(t,k)}\|^2 + \beta\|Q^{(t)}\|^2.$$

(61)

For a sufficiently small learning rate $\eta$, term (A) is non-negative. Thus, to upper bound $\|\nabla F(w^{(t)})\|^2$, it suffices to keep track of $\|\nabla F(\bar{w}^{(t-1)}) - g^{(t,k)}\|^2$. Conditional on $\bar{w}^{(t-1)}$, take expectation on both sides of (54) and we have

$$\frac{\eta K}{2}\mathbb{E}[\|\nabla F(\bar{w}^{(t-1)})\|^2] \leq \mathbb{E}\big[F(\bar{w}^{(t-1)}) - F(\bar{w}^{(t)}) - (\frac{\eta}{2n} - \frac{\beta\eta^2 K}{n}) \sum_{i=1}^{n} \sum_{k=0}^{K-1} \|g_i^{(t,k)}\|^2$$

$$+ \frac{\eta}{2n} \sum_{i=1}^{n} \sum_{k=0}^{K-1} \|\nabla F(\bar{w}^{(t-1)}) - g_i^{(t,k)}\|^2 + \beta\|Q^{(t)}\|^2\big],$$

(62)

since $\mathbb{E}[B_t] = nq$.

By AM-GM inequality again,

$$\sum_{i=1}^{n} \|\nabla F(\bar{w}^{(t-1)}) - g_i^{(t,k)}\|^2$$

$$\leq 2 \sum_{i=1}^{n} \left( \|\nabla F(\bar{w}^{(t-1)}) - \nabla f_i(\bar{w}^{(t-1)})\|^2 + \|\nabla f_i(\bar{w}^{(t-1)}) - \nabla f_i(w_i^{(t,k)})\|^2 \right)$$

$$\leq 2\left(n\tau + \beta^2 \sum_{i=1}^{n} \|\bar{w}^{(t-1)} - w_i^{(t,k)}\|^2\right)$$

$$= 2\left(n\tau + \beta^2 \eta^2 \sum_{i=1}^{n} \|\sum_{l=0}^{k-1} g_i^{(t,l)}\|^2\right) \leq 2\left(n\tau + \beta^2 \eta^2 k \sum_{i=1}^{n} \sum_{l=0}^{k-1} \|g_i^{(t,l)}\|^2\right). \tag{63}$$

Plugging (63), which suggests that

$$\frac{\eta}{2n} \sum_{i=1}^{n} \sum_{k=0}^{K-1} \|\nabla F(\bar{w}^{(t-1)}) - g_i^{(t,k)}\|^2 \leq \eta\left(\tau K + \frac{\beta^2 \eta^2 K^2}{n} \sum_{i=1}^{n} \sum_{k=0}^{K-1} \|g_i^{(t,k)}\|^2\right),$$

back to (62), we have that

$$\frac{\eta K}{2} \mathbb{E}[\|\nabla F(\bar{w}^{(t-1)})\|^2] \leq \mathbb{E}\left[ F(\bar{w}^{(t-1)}) - F(\bar{w}^{(t)}) - \left(\frac{\eta}{2n} - \frac{\beta \eta^2 K}{n} - \frac{\beta^2 \eta^3 K^2}{n}\right) \sum_{i=1}^{n} \sum_{k=0}^{K-1} \|g_i^{(t,k)}\|^2 \right.$$

$$\left. + \eta\tau K + \beta\|Q^{(t)}\|^2 \right], \tag{64}$$

Therefore, when $\frac{\eta}{2n} - \frac{\beta \eta^2 K}{n} - \frac{\beta^2 \eta^3 K^2}{n} \geq 0$, which requires that $\eta \leq \frac{1}{2\beta K}$, we have

$$\mathbb{E}[\|\nabla F(\bar{w}^{(t-1)})\|^2] \leq 2 \cdot \mathbb{E}\left[ \frac{F(\bar{w}^{(t-1)}) - F(\bar{w}^{(t)})}{\eta K} + \tau + \frac{\beta}{\eta K}\|Q^{(t)}\|^2 \right]. \tag{65}$$

Now, we sum up (65) both sides for $t = 1, 2, \cdots, T$ and average them, we have that

$$\mathbb{E}\left[ \frac{\sum_{t=1}^{T} \|\nabla F(\bar{w}^{(t-1)})\|^2}{T} \right] \leq 2 \cdot \mathbb{E}\left[ \frac{F(\bar{w}^{(t-1)})}{\eta T K} + \tau + \frac{\sum_{t=1}^{T} \beta \mathbb{E}[\|Q^{(t)}\|^2]}{\eta T K} \right]. \tag{66}$$

## D  PROOF OF THEOREM 1: UTILITY OF DP-LSGD IN GENERAL CONVEX OPTIMIZATION

We first focus on the clipped local update $\mathcal{CP}(\Delta w_i^{(t)}, c) = \mathcal{CP}(w_i^{(t,K)} - \bar{w}^{(t-1)}, c)$ in the $t$-th phase if the $i$-th sample gets selected. Since the local update before clipping is essentially the sum of gradient scaled by the learning rate $-\eta$, therefore,

$$\mathcal{CP}(w_i^{(t,K)} - \bar{w}^{(t-1)}, c) = \mathcal{CP}(-\eta \sum_{k=0}^{K-1} \nabla f_i(w_i^{(t,k)}), c) = -\eta_i^{(t)} \sum_{k=0}^{K-1} \nabla f_i(w^{(t,k)}), \quad (67)$$

where $\eta_i^{(t)} = \eta \cdot \min\{1, \frac{c}{\|\sum_{k=0}^{K-1} \nabla f_i(w_i^{(t,k)})\|}\}$ is determined by the clipping threshold, and thus $\eta_i^{(t)} \le \eta$. Based on Definition 4,

$$\eta - \eta_i^{(t)} = \eta \cdot (1 - \frac{c}{c + \mathbf{1}(\|\Delta w_i^{(t)}\| > c) \cdot (\|\Delta w_i^{(t)}\| - c)}) = \eta \cdot \frac{\Psi_i^{(t)}}{c + \Psi_i^{(t)}}, \quad (68)$$

where $\Psi_i^{(t)} = \max\{0, \|\Delta w_i^{(t)}\| - c\}$ represents the incremental norm of the local update from the $i$-th sample in the $t$-th phase. For simplicity, we will use $\Delta\Psi_i^{(t)}$ to denote $\frac{\Psi_i^{(t)}}{c+\Psi_i^{(t)}}$.

Now, we consider two virtual sequences:

a) $w_i'^{(t,0)} = \bar{w}^{(t-1)}$ and $w_i'^{(t,k)} = w_i'^{(t,k-1)} - \eta_i^{(t)} \nabla f_i(w_i^{(t,k-1)})$, which represents a sequence of iterates based on the gradients $\nabla f_i(w^{(t,k-1)})$ but scaled by $\eta_i^{(t)}$ instead of constant $\eta$ for each $i$;

b) We use $\hat{w}^{(t,k)} = \frac{1}{nq} \cdot \sum_{i=1}^{n} \mathbf{1}_i^{(t)} \cdot w_i'^{(t,k)}$ to represent the average of $w_i'^{(t,k)}$ for those indices $i$ selected in the $t$-th phase. Here, $\mathbf{1}_i^{(t)} = 1$ iff the $i$-th sample is selected in the $t$-th phase. Similarly, we define $\tilde{w}^{(t,k)} = \frac{1}{n} \cdot w_i'^{(t,k)}$ to be the average of all $w_i'^{(t,k)}$ for $i = 1, 2, \cdots, n$. It is not hard to observe that $\tilde{w}^{(t,K)} = \bar{w}^{(t-1)} + \mathcal{CP}(\Delta w_i^{(t)}, c)$, and consequently conditional on $\bar{w}^{(t-1)}$, $\mathbb{E}[\bar{w}^{(t)}] = \mathbb{E}[\hat{w}^{(t,K)}] = \tilde{w}^{(t,K)}$ since the independent DP noise satisfies that $\mathbb{E}[Q^{(t)}] = 0$.

In the following, we unravel $\|\tilde{w}^{(t,k)} - u\|^2$ for arbitrary $u$ and obtain

$$\|\hat{w}^{(t,k)} - u\|^2$$

$$= \|\hat{w}^{(t,k-1)} - \sum_{i=1}^{n} \frac{\eta_i^{(t)} \cdot \mathbf{1}_i^{(t)} \cdot \nabla f_i(w_i^{(t,k-1)})}{nq} - u\|^2$$

$$= \|\hat{w}^{(t,k-1)} - u\|^2 - \frac{2}{nq} \cdot \sum_{i=1}^{n} \eta_i^{(t)} \mathbf{1}_i^{(t)} \langle \tilde{w}^{(t,k-1)} - u, \nabla f_i(w_i^{(t,k-1)}) \rangle + \|\frac{\sum_{i=1}^{n} \eta_i^{(t)} \mathbf{1}_i^{(t)} \nabla f_i(w_i^{(t,k-1)})}{nq}\|^2.$$

$$(69)$$

We first work on the last term of (69). With the fact that $\eta_i^{(t)} \le \eta$, conditional on $\bar{w}^{(t-1)}$,

$$\mathbb{E}[\|\frac{\sum_{i=1}^{n} \eta_i^{(t)} \mathbf{1}_i^{(t)} \nabla f_i(w_i^{(t,k-1)})}{nq}\|^2]$$

$$= \mathbb{E}[\|\frac{\sum_{i=1}^{n} \eta_i^{(t)} \mathbf{1}_i^{(t)} \nabla f_i(w_i^{(t,k-1)})}{nq} - \frac{\sum_{i=1}^{n} \eta_i^{(t)} \nabla f_i(w_i^{(t,k-1)})}{n} + \frac{\sum_{i=1}^{n} \eta_i^{(t)} \nabla f_i(w_i^{(t,k-1)})}{n}\|^2]$$

$$\le 2 \cdot \mathbb{E}[\|\frac{\sum_{i=1}^{n} \eta_i^{(t)} (\mathbf{1}_i^{(t)} - q) \nabla f_i(w_i^{(t,k-1)})}{nq}\|^2] + 2 \cdot \|\frac{\sum_{i=1}^{n} \eta_i^{(t)} \nabla f_i(w_i^{(t,k-1)})}{n}\|^2$$

$$\le \frac{2(q - q^2) \sum_{i=1}^{n} \|\eta_i^{(t)} \nabla f_i(w_i^{(t,k-1)})\|^2}{(nq)^2} + \frac{2 \sum_{i=1}^{n} \|\eta_i^{(t)} \nabla f_i(w_i^{(t,k-1)})\|^2}{n}$$

$$\le \frac{4\eta^2 \sum_{i=1}^{n} \|\nabla f_i(w_i^{(t,k-1)})\|^2}{n}$$

$$(70)$$

which can be further bounded via Lemma 2 as

$$4\eta^2\Big(\frac{3\beta^2\sum_{i=1}^n\|w_i^{(t,k-1)}-\tilde{w}^{(t,k-1)}\|^2}{n}+\min\{6\beta F(\tilde{w}^{(t,k-1)})-F(w^*),3\beta^2\|\tilde{w}^{(t,k-1)}-w^*\|^2\}+3\tau\Big). \tag{71}$$

Now, we move our focus to the second term of (69). Still, with a similar reasoning as Lemma 3,

$$\mathbb{E}\Big[\frac{-2}{nq}\cdot\sum_{i=1}^n\mathbf{1}_i^{(t)}\eta_i^{(t)}\langle\tilde{w}^{(t,k-1)}-u,\nabla f_i(w_i^{(t,k-1)})\rangle\Big]$$

$$=\Big[\frac{-2}{n}\cdot\sum_{i=1}^n\eta(1-\Delta\Psi_i^{(t)})\langle\tilde{w}^{(t,k-1)}-u,\nabla f_i(w_i^{(t,k-1)})\rangle\Big]$$

$$\leq\frac{2}{n}\sum_{i=1}^n\eta(1-\Delta\Psi_i^{(t)})\big(f_i(u)-f_i(\tilde{w}^{(t,k-1)})+\frac{\beta}{2}\|w_i^{(t,k-1)}-\tilde{w}^{(t,k-1)}\|^2\big)$$

$$\leq2\eta\big(F(u)-F(\tilde{w}^{(t,k-1)})+\frac{\beta}{2n}\cdot\sum_{i=1}^n(1-\Delta\Psi_i^{(t)})\|w^{(t,k-1)}-\tilde{w}^{(t,k-1)}\|^2\big)$$

$$\quad-\frac{2}{n}\cdot\sum_{i=1}^n\eta\Delta\Psi_i^{(t)}\big(F(u)-F(\tilde{w}^{(t,k-1)})\big)+\sum_{i=1}^n\frac{2}{n}\big(\eta\Delta\Psi_i^{(t)}\big)\cdot2\gamma$$

$$\leq2\eta(1-\frac{\sum_{i=1}^n\Delta\Psi_i^{(t)}}{n})\big(F(u)-F(\tilde{w}^{(t,k-1)})\big)+\big(\frac{\beta\eta}{n}\sum_{i=1}^n\|w_i^{(t,k-1)}-\tilde{w}^{(t,k-1)}\|^2\big)+\frac{4\eta\gamma\sum_{i=1}^n\Delta\Psi_i^{(t)}}{n}. \tag{72}$$

In the fourth line of (72), we use the $\gamma$-similarity assumption from Assumption 2. In the following, we will use $\Delta\bar{\Psi}^{(t)}=\frac{\sum_{i=1}^n\Delta\Psi_i^{(t)}}{n}$ for simplicity.

Next, we work on the upper bound of $\sum_{i=1}^n\|w_i^{(t,k-1)}-\tilde{w}^{(t,k-1)}\|^2$. Similar to Lemma 4,

$$\sum_{i=1}^n\|\tilde{w}^{(t,k-1)}-w_i^{(t,k-1)}\|^2$$

$$=\sum_{i=1}^n\|\frac{\sum_{l=0}^{k-1}\sum_{j=1}^n\eta_j^{(t)}\nabla f_j(w_j^{(t,l)})}{n}-\eta\cdot\sum_{l=0}^{k-1}\nabla f_i(w_i^{(t,l)})\|^2$$

$$\leq2\sum_{i=1}^n\Big(\eta^2\|\frac{\sum_{l=0}^{k-1}\sum_{j=1}^n(\nabla f_j(w_j^{(t,l)})-\nabla f_i(w_i^{(t,l)}))}{n}\|^2+\|\frac{\sum_{l=0}^{k-1}\sum_{j=1}^n(\eta-\eta_j^{(t)})\nabla f_j(w_j^{(t,l)})}{n}\|^2\Big) \tag{73}$$

For the first term in (73), we have studied it in Lemma 4, where once $\eta^2<\frac{\beta^2}{24K^2}$,

$$\sum_{i=1}^n\|\eta\cdot\frac{\sum_{l=0}^{k-1}\sum_{j=1}^n\nabla f_j(w_j^{(t,l)})}{n}-\eta\cdot\sum_{l=0}^{k-1}\nabla f_i(w_i^{(t,l)})\|^2\leq4\eta^2k^2n\tau. \tag{74}$$

Plugging (74) back to (73), since $(\eta-\eta_j^{(t)})^2\leq\eta^2$, and we apply the similar decomposition trick used in (71), we have that

$$\sum_{i=1}^n\frac{\|\tilde{w}^{(t,k-1)}-w_i^{(t,k-1)}\|^2}{n}\leq8\eta^2k^2n\tau+\frac{1}{n}\cdot\frac{2k\eta^2\sum_{l=0}^{k-1}\sum_{i=1}^n\|\nabla f_i(w_i^{(t,l)})\|^2}{n}$$

$$\leq8\eta^2k^2\tau$$

$$\quad+\frac{6k\eta^2}{n}\sum_{l=0}^{k-1}\big(\beta^2\|\tilde{w}^{(t,l)}-w_i^{(t,l)}\|^2+\min\big\{2\beta\big(F(\tilde{w}^{(t,l)})-F(w^*)\big),\beta^2\|\tilde{w}^{(t,l)}-w^*\|^2\big\}+\tau\big)$$

$$\leq14\eta^2k^2\tau+\frac{6k\eta^2}{n}\sum_{l=0}^{k-1}\big(\beta^2\|\tilde{w}^{(t,l)}-w_i^{(t,l)}\|^2+\min\big\{2\beta\big(F(\tilde{w}^{(t,l)})-F(w^*)\big),\beta^2\|\tilde{w}^{(t,l)}-w^*\|^2\big\}\big), \tag{75}$$

given that $n \geq 1$. Thus, when $\eta$ is selected small enough such that $\eta \leq \min\{\frac{\sqrt{n}}{\sqrt{30}K\beta}, \frac{1}{\sqrt{6}K}\}$, for any $k_0 \leq K$, by induction it is not hard to verifiy that

$$
\frac{\sum_{i=1}^{n} \|w_i^{(t,k_0-1)} - \tilde{w}^{(t,k_0-1)}\|^2}{n}
$$
$$
\leq 15\eta^2 k_0^2\tau + \frac{12\eta^2 k_0}{n}\Big(\sum_{l=0}^{k_0-1} \min\big\{2\beta\big(F(\tilde{w}^{(t,l)}) - F(w^*)\big), \beta^2\|\tilde{w}^{(t,l)} - w^*\|^2\big\}\Big). \tag{76}
$$

Now, we put (71), (72) and (76) together, and go back to (69)

$$
[\eta(1 - \Delta\bar{\Psi}^{(t)})\big(F(\tilde{w}^{(t,k-1)}) - F(u)\big)] \leq \mathbb{E}[\|\hat{w}^{(t,k-1)} - u\|^2 - \|\hat{w}^{(t,k)} - u\|^2] + 4\eta\gamma\Delta\bar{\Psi}^{(t)}
$$
$$
+ (12\eta^2\beta^2 + \beta\eta)\big(15\eta^2 k^2\tau + \frac{12\eta^2 k}{n}\big(\sum_{l=0}^{k-1} \min\big\{2\beta\big(F(\tilde{w}^{(t,l)}) - F(w^*)\big), \beta^2\|\tilde{w}^{(t,l)} - w^*\|^2\big\}\big)\big)
$$
$$
+ 12\eta^2 \min\big\{2\beta\big(F(\tilde{w}^{(t,k-1)}) - F(w^*)\big), \beta^2\|\tilde{w}^{(t,l)} - w^*\|^2\big\} + 12\eta^2\tau \tag{77}
$$

When $\eta$ is small enough such that $12\eta^2\beta^2 + \beta\eta \leq 2\beta\eta$, (77) can be simplified as

$$
[\eta(1 - \Delta\bar{\Psi}^{(t)})\big(F(\tilde{w}^{(t,k-1)}) - F(u)\big)] \leq \mathbb{E}[\|\hat{w}^{(t,k-1)} - u\|^2 - \|\hat{w}^{(t,k)} - u\|^2] + 4\eta\gamma\Delta\bar{\Psi}^{(t)}
$$
$$
+ (10K^2\beta\eta^3 + 12\eta^2)\tau + \frac{24K\beta\eta^3}{n}\sum_{l=0}^{k-1} \min\big\{2\beta\big(F(\tilde{w}^{(t,l)}) - F(w^*)\big), \beta^2\|\tilde{w}^{(t,l)} - w^*\|^2\big\}
$$
$$
+ 12\eta^2 \min\big\{2\beta\big(F(\tilde{w}^{(t,k-1)}) - F(w^*)\big), \beta^2\|\tilde{w}^{(t,l)} - w^*\|^2\big\}. \tag{78}
$$

The remainder of the proof is almost the same as that for Theorem 1. On one hand, it is noted that

$$
1 - \Delta\bar{\Psi}^{(t)} = \sum_{i=1}^{n} \frac{1}{n} \cdot \frac{c}{c + \Psi_i^{(t)}} \geq \frac{c}{c + \frac{\Psi_i^{(t)}}{n}}, \tag{79}
$$

since $1/(1 + x)$ is convex regarding $x$. Therefore, $\mathbb{E}[(1 - \Delta\bar{\Psi}^{(t)})] \geq \frac{c}{c+\mathcal{B}}$ and $\mathbb{E}[\Delta\bar{\Psi}^{(t)}] \leq \frac{\mathcal{B}}{c+\mathcal{B}}$ by Assumption 5 that $\mathbb{E}[\frac{\sum_{i=1}^{n}\Psi_i^{(t)}}{n}] \leq \mathcal{B}$.

Therefore, for sufficiently small $\eta = O(n/K^2)$ such that $24\eta^2\beta + \frac{48K^2\beta^2\eta^3}{n} \leq \frac{c\eta}{2(c+\mathcal{B})}$, summing up both sides of (77) for $k = 1, 2, \cdots, K$ and $t = 1, 2, \cdots, T$ with $u = w^*$, and take the zero-mean independent DP noise into accountant where $\bar{w}^{(t)} = \hat{w}^{(t,K)} + Q^{(t)}$, we have

$$
\mathbb{E}\Big[\frac{\sum_{t=1}^{T} \sum_{k=1}^{K-1} \frac{c}{2(c+\mathcal{B})}\big(F(\tilde{w}^{(t,k-1)}) - F(w^*)\big)}{TK}\Big]
$$
$$
\leq \frac{\|\bar{w}^{(0)} - w^*\|^2}{TK\eta} + (30K^2\beta\eta^2 + 12\eta)\tau + \frac{4\gamma\mathcal{B}}{c+\mathcal{B}} + \frac{\sigma^2 d}{K\eta}. \tag{80}
$$

To obtain the convergence guarantee of $\bar{w}^{(T)}$, we similarly imagine a virtual step where we implement one additional full gradient descent using the entire set and we have that

$$
\|\tilde{w}^{(T+1,1)} - u\|^2 = \|\bar{w}^{(T)} - u - \eta \cdot \frac{\sum_{i=1}^{n} \nabla f_i(\tilde{w}^{(T,K)})}{n}\|^2
$$
$$
\leq \|\bar{w}^{(T)} - u\|^2 - 2\eta\big(F(\bar{w}^{(T)}) - F(u)\big) + \eta^2\|\nabla F(\bar{w}^{(T)}) - \nabla F(w^*)\|^2
$$
$$
\leq \|\bar{w}^{(T)} - w^*\|^2 - 2\eta\big(F(\bar{w}^{(T)}) - F(u)\big) + \eta^2 \min\{\beta^2\|\bar{w}^{(T)} - w^*\|^2, 2\beta(F(\bar{w}^T) - F(w^*))\}\}. \tag{81}
$$

Therefore, for small enough $\eta$, such that $\eta - \eta^2\beta > 0.5\eta$, we combine (80) and (81) with $u = w^*$, and have

$$
\mathbb{E}\Big[\frac{\sum_{t=1}^{T} \sum_{k=1}^{K} \frac{c}{2(c+\mathcal{B})}\big(F(\tilde{w}^{(t,k-1)}) - F(w^*)\big) + \frac{\mathcal{B}}{2(c+\mathcal{B})}\big(F(\bar{w}^{(T)}) - F(w^*)\big)}{TK + 1}\Big]
$$
$$
\leq \frac{\|\bar{w}^{(0)} - w^*\|^2}{(TK + 1)\eta} + (30K^2\beta\eta^2 + 12\eta)\tau + \frac{4\gamma\mathcal{B}}{c+\mathcal{B}} + \frac{\sigma^2 d}{K\eta}. \tag{82}
$$

Similarly, it is noted that conditional on $\bar{w}^{(t-1)}$, we still have that

$$\mathbb{E}[\|\hat{w}^{(t,k)} - u\|^2] = \mathbb{E}[\|\hat{w}^{(t,k)} - \tilde{w}^{(t,k)}\|^2] + \|\tilde{w}^{(t,k)} - u\|^2, \tag{83}$$

and for $\mathbb{E}[\|\hat{w}^{(t,k)} - \tilde{w}^{(t,k)}\|^2]$ for any $t$ and $k$, we use $\tilde{w}'^{(t,k)} = \frac{1}{n} \cdot \sum_{i=1}^n w_i^{(t,k)}$,

$$
\begin{aligned}
\mathbb{E}[\|\hat{w}^{(t,k)} - \tilde{w}^{(t,k)}\|^2] &= \mathbb{E}[\|(\hat{w}^{(t,k)} - \bar{w}^{(t-1)}) - (\tilde{w}^{(t,k)} - \bar{w}^{(t-1)})\|^2] \\
&= \mathbb{E}[\|\sum_{i=1}^n \frac{\eta_i^{(t)}}{\eta} \cdot \frac{\mathbf{1}_i^{(t)} - q}{nq} \cdot \sum_{l=0}^{k-1} \nabla f_i(w_i^{(t,l)})\|^2] \leq \frac{k}{n^2 q} \sum_{i=1}^n \sum_{l=0}^{k-1} \|\nabla f_i(w_i^{(t,l)})\|^2,
\end{aligned}
\tag{84}
$$

since $\eta_i^{(t)} \leq \eta$. Therefore, by (24), we also have that

$$\mathbb{E}[\|\hat{w}^{(t,k)} - \tilde{w}^{(t,k)}\|^2] \leq \frac{3K\eta^2}{nq}\left(4\beta^2 K^3 \tau \eta^2 + K\tau + \sum_{l=0}^{k-1} \beta^2 \|\tilde{w}^{(t,l)} - w^*\|^2\right) \tag{85}$$

Now, using (71) and (83), (78) can be rewritten as

$$
\begin{aligned}
&[\eta(1 - \Delta\bar{\Psi}^{(t)})\left(F(\tilde{w}^{(t,k-1)}) - F(u)\right)] \\
&\leq \mathbb{E}[\|\tilde{w}^{(t,k-1)} - u\|^2 - \|\tilde{w}^{(t,k)} - u\|^2 + \|\tilde{w}^{(t,k-1)} - \hat{w}^{(t,k-1)}\|^2 - \|\tilde{w}^{(t,k)} - \hat{w}^{(t,k)}\|] \\
&+ \frac{\eta^2 K}{nq} \sum_{l=1}^k \left(\frac{3\beta^2 \sum_{i=1}^n \|w_i^{(t,l-1)} - \tilde{w}^{(t,k-1)}\|^2}{n} + \min\{6\beta F(\tilde{w}^{(t,k-1)}) - F(w^*), 3\beta^2 \|\tilde{w}^{(t,k-1)} - w^*\|^2\} + 3\tau\right) \\
&+ (10K^2\beta\eta^3 + 12\eta^2)\tau + \frac{24K\beta\eta^3}{n} \sum_{l=0}^{k-1} \min\left\{2\beta\left(F(\tilde{w}^{(t,l)}) - F(w^*)\right), \beta^2 \|\tilde{w}^{(t,l)} - w^*\|^2\right\} \\
&+ 12\eta^2 \min\left\{2\beta\left(F(\tilde{w}^{(t,k-1)}) - F(w^*)\right), \beta^2 \|\tilde{w}^{(t,l)} - w^*\|^2\right\}.
\end{aligned}
\tag{86}
$$

On the other hand, if we select $u = \tilde{w}^{(t_0,k_0)}$ for some $t_0 \in [1 : T]$ and $k_0 \in [0, K-1]$ in (86), when $K^2 = O(nq)$,

$$
\begin{aligned}
&\mathbb{E}\left[\frac{\sum_{(t,k)\in\mathcal{C}} \frac{c}{2(c+\mathcal{B})}\left(F(\tilde{w}^{(t,k)}) - F(\tilde{w}^{(t_0,k_0)})\right) + \frac{c}{2(c+\mathcal{B})}(F(\bar{w}^T) - F(\tilde{w}^{(t_0,k_0)}))}{(T - t_0 + 1)K - k_0 + 1}\right] \\
&\leq O(1) \cdot \left\{\frac{\frac{3K\eta}{nq}\left(4\beta^2 K^3 \tau \eta^2 + K\tau + \sum_{l=0}^{k-1} \beta^2 \|\tilde{w}^{(t,l)} - w^*\|^2\right)}{(T - t_0 + 1)K - k_0 + 1}\right. \\
&\frac{K\beta^3\eta^2}{n}\left(\frac{\sum_{(t,k)\in\mathcal{C}} \sum_{l=0}^{K-1} \mathbb{E}[\|\tilde{w}^{(t,l)} - w^*\|^2]}{(T - t_0 + 1)K - k_0 + 1}\right) + (K^2\beta\eta^2 + \eta)\tau \\
&\left. + \frac{\gamma\mathcal{B}}{(c+\mathcal{B})} + \frac{\sigma^2 d}{\eta} + \eta\beta^2 \frac{\sum_{(t,k)\in\mathcal{C}} \mathbb{E}[\|\tilde{w}^{(t,k-1)} - w^*\|^2] + \mathbb{E}[\|\bar{w}^{(T)} - w^*\|^2]}{(T - t_0 + 1)K - k_0 + 1}\right\},
\end{aligned}
\tag{87}
$$

where $\mathcal{C} = \left((t_0, k), k = k_0, \cdots, K-1\right) \cup \left((t,k), t = t_0 + 1, \cdots, T, k = 0, \cdots, K-1\right)$. In the following, we may apply a similar reasoning as Lemma 5 to derive the following results.

**Lemma 7.** *Provided sufficiently small $\eta = o(1/K)$, for any $t \in [1 : T]$ and $k \in [0 : K-1]$*

$$\mathbb{E}[\|\tilde{w}^{(t,k)} - w^*\|^2] = O\left(\|\bar{w}^{(0)} - w^*\|^2 + TK\left(\eta\gamma\frac{\mathcal{B}}{c+\mathcal{B}} + \eta^3 K^2 \tau + \eta^2 \tau + \frac{K\tau\eta^2}{nq}\right) + T\sigma^2 d\right).$$

By Lemma (7),

$$
\begin{aligned}
&\frac{24K\beta^3\eta^2}{n} \cdot \frac{\sum_{(t,k)\in\mathcal{C}} \sum_{l=0}^{K-1} \mathbb{E}[\|\tilde{w}^{(t,l)} - w^*\|^2] + \mathbb{E}[\|\bar{w}^{(T)} - w^*\|^2]}{(T - t_0 + 1)K - k_0} \\
&\leq \frac{K^2\beta^3\eta^2}{n} \cdot O\left(\|\bar{w}^{(0)} - w^*\|^2 + TK\left(\eta\gamma\frac{\mathcal{B}}{c+\mathcal{B}} + \eta^3 K^2 \tau + \eta^2 \tau + \frac{K\tau\eta^2}{nq}\right) + T\sigma^2 d\right).
\end{aligned}
\tag{88}
$$

On the other hand, we have

$$12\eta\beta^2 \frac{\sum_{t=t_0}^{T}\sum_{k=k_0+1}^{K-1}\mathbb{E}[\|\tilde{w}^{(t,k-1)}-w^*\|^2]}{(T-t_0+1)K-k_0}.$$

$$\leq \eta \cdot O\big(\|\bar{w}^{(0)}-w^*\|^2 + TK\big(\eta\gamma\frac{\mathcal{B}}{c+\mathcal{B}} + \eta^3 K^2\tau + \eta^2\tau + \frac{K\tau\eta^2}{nq}\big) + T\sigma^2 d\big). \tag{89}$$

Now, we can apply the last iterate trick in Lemma 6. Let $y_j = \frac{c}{2(c+\mathcal{B})}\mathbb{E}[\big(F(\tilde{w}^{(t,k)})-F(w^*)\big)]$ for $j=(t-1)K+k+1$ for $t=1,2,\cdots,T$ and $k=0,1,\cdots,K-1$, and $y_{TK+1}=\frac{c}{2(c+\mathcal{B})}\mathbb{E}[F(\bar{w}^{(T)})-F(w^*)]$.

$$\begin{aligned}
y_{TK+1} &= \mathbb{E}[\frac{c}{2(c+\mathcal{B})}(F(\bar{w}^{(T)})-F(w^*))]\\
&= \frac{\sum_{j=1}^{TK+1}y_j}{TK+1} + \sum_{j=1}^{TK}\frac{1}{j+1}\cdot\frac{\sum_{l=TK+1-j}^{TK+1}(y_l-y_{TK+1-j})}{j}\\
&\leq \tilde{O}\big((\eta+\frac{\eta^2 K^2}{n}+\frac{K^2\eta}{nq}+\frac{1}{TK\eta})\cdot\|\bar{w}^{(0)}-w^*\|^2\\
&\quad + TK(\frac{K^2\eta^2}{n}+\frac{K^2\eta}{nq}+\eta)\cdot\big((1+K^2\eta+\frac{K}{nq})\eta^2\tau+\eta\frac{\gamma\mathcal{B}}{c+\mathcal{B}}\big)+\frac{K\eta}{nq}\big(\beta^2 K^3\tau\eta^2+K\tau\big)\\
&\quad + (\frac{K^2\eta}{nq}+\frac{TK^2\eta^2}{n}+T\eta+1/\eta)\sigma^2 d\big)\\
&= \tilde{O}\big((\frac{1}{\sqrt{TK}}+\frac{K}{nT})\|\bar{w}^{(0)}-w^*\|^2 + (\frac{K}{nT}+\frac{1}{\sqrt{TK}})(1+\frac{K^{3/2}}{\sqrt{T}}+\frac{K}{nq})\tau+(K^2\eta^3+\eta)\tau\\
&\quad + (\frac{K^{3/2}}{\sqrt{T}n}+1)\frac{\gamma\mathcal{B}}{c+\mathcal{B}}+\sqrt{TK}\sigma^2 d\big)\\
&= \tilde{O}\big(\frac{\|\bar{w}^{(0)}-w^*\|^2}{\sqrt{TK}}+(\frac{1}{\sqrt{TK}}+\frac{K}{T})\tau+\frac{\gamma\mathcal{B}}{c+\mathcal{B}}+\sqrt{TK}\sigma^2 d\big).
\end{aligned} \tag{90}$$

when we select $\eta=O(1/\sqrt{TK})$, $K=O(nq)$ and $K=O(T)$. This completes the proof.

### D.1 Proof of Lemma 7

From (69), by letting $u=w^*$, given $\bar{w}^{(t-1)}$, we have that

$$\begin{aligned}
&\|\tilde{w}^{(t,k)}-u\|^2\\
&= \|\tilde{w}^{(t,k-1)}-\sum_{i=1}^{n}\frac{\eta_i^{(t)}\cdot\nabla f_i(w_i^{(t,k-1)})}{n}-w^*\|^2\\
&= \|\tilde{w}^{(t,k-1)}-w^*\|^2 - \frac{2}{n}\cdot\sum_{i=1}^{n}\eta_i^{(t)}\langle\tilde{w}^{(t,k-1)}-w^*,\nabla f_i(w_i^{(t,k-1)})\rangle + \|\frac{\sum_{i=1}^{n}\eta_i^{(t)}\nabla f_i(w_i^{(t,k-1)})}{n}\|^2.
\end{aligned} \tag{91}$$

By (72) and (70), (91) can be further bounded by

$$\|\tilde{w}^{(t,k)} - w^*\|^2$$

$$= \|\tilde{w}^{(t,k-1)} - w^*\|^2 + 2\eta(1 - \Delta\bar{\Psi}^{(t)})\big(F(w^*) - F(\tilde{w}^{(t,k-1)})\big) + \big(\frac{\beta\eta}{n}\sum_{i=1}^n \|w_i^{(t,k-1)} - \tilde{w}^{(t,k-1)}\|^2\big)$$

$$+ 4\eta\gamma\Delta\bar{\Psi}^{(t)} + \eta^2\big(\frac{3\beta^2\sum_{i=1}^n \|w_i^{(t,k-1)} - \tilde{w}^{(t,k-1)}\|^2}{n} + 6\beta(F(\tilde{w}^{(t,k-1)}) - F(w^*)) + 3\tau\big)$$

$$\leq \|\tilde{w}^{(t,k-1)} - w^*\|^2 - \big(2\eta(1 - \Delta\bar{\Psi}^{(t)}) - 6\beta\eta^2\big)\big(F(\tilde{w}^{(t,k-1)}) - F(w^*)\big)$$

$$+ (\eta\beta + 3\eta^2\beta^2)\frac{\sum_{i=1}^n \|w_i^{(t,k-1)} - \tilde{w}^{(t,k-1)}\|^2}{n} + 4\eta\gamma\Delta\bar{\Psi}^{(t)} + 3\eta^2\tau$$

$$\leq \|\tilde{w}^{(t,k-1)} - w^*\|^2 - \big(2\eta(1 - \Delta\bar{\Psi}^{(t)}) - 6\beta\eta^2\big)\big(F(\tilde{w}^{(t,k-1)}) - F(w^*)\big)$$

$$+ (\eta\beta + 3\eta^2\beta^2)\big(15\eta^2 k^2\tau + \frac{12\eta^2 k}{n}\big(\sum_{l=0}^{k-1}\beta\big(F(\tilde{w}^{(t,l)}) - F(w^*)\big)\big) + 4\eta\gamma\Delta\bar{\Psi}^{(t)} + 3\eta^2\tau.$$

$$(92)$$

On the other hand, as for $\|\bar{w}^{(t+1)} - w^*\|$, we have that

$$\mathbb{E}[\|\bar{w}^{(t)} - w^*\|^2] = \mathbb{E}[\|\bar{w}^{(t)} - \tilde{w}^{(t,K)}\|^2] + \mathbb{E}[\|\tilde{w}^{(t,K)} - w^*\|^2]$$

$$= \mathbb{E}[\|\frac{\sum_{k=1}^K \sum_{i=1}^n (1_i^{(1)} - q)\eta_i^{(t)}\nabla f_i(w_i^{(t,k-1)})}{nq}\|^2] + \mathbb{E}[\|\tilde{w}^{(t,K)} - w^*\|^2] + \sigma^2 d$$

$$\leq \frac{K\eta^2\sum_{k=1}^K \sum_{i=1}^n \|\nabla f_i(w_i^{(t,k-1)})\|^2}{n^2 q} + \mathbb{E}[\|\tilde{w}^{(t,K)} - w^*\|^2] + \sigma^2 d$$

$$\leq \frac{3K\eta^2\sum_{k=1}^K \big\{\sum_{i=1}^n \big(\beta^2\|w_i^{(t,k-1)} - \tilde{w}^{(t,k-1)}\|^2\big) + 2\beta n(F(\tilde{w}^{(t,k-1)}) - F(w^*)) + n\tau\big\}}{n^2 q}$$

$$+ \mathbb{E}[\|\tilde{w}^{(t,K)} - w^*\|^2] + \sigma^2 d$$

$$= O\big(\|\bar{w}^{(0)} - w^*\|^2 + tK\big(\eta\gamma\frac{\mathcal{B}}{c + \mathcal{B}} + (\eta^2 + \eta^3 K^2)\tau + \frac{K\tau\eta^2}{nq}\big) + t\sigma^2 d\big).$$

$$(93)$$

for sufficiently small $\eta = o(1/K)$ and $K = O(nq)$. Thus, with the above reasoning, we consider $t = T$ and $k = K$, and then we obtain a global upper bound.

# E   UTILITY OF DP-LSGD IN STRONGLY CONVEX OPTIMIZATION

**Theorem 5.** *For an arbitrary objective loss function $F(w) = \frac{1}{n}\cdot\sum_{i=1}^n f_i(w)$ where $f_i(w)$ is $\lambda$-strongly-convex and $\beta$-smooth, when $\eta < \min\{1/\beta, 2/(\beta + \lambda)\}$, Algorithm 1 with clipped local update (3) ensures that*

$$\mathbb{E}[\|\bar{w}^{(T)} - w^*\|^2] \leq \big(1 - (\eta\lambda)^2\big)^{TK}\|\bar{w}^{(0)} - w^*\|^2 + \frac{4(1 + \eta\lambda)^K\cdot\big(\frac{c^2}{nq} + \mathcal{B}^2 + \eta^2\tau K^2 + \sigma^2 d\big)}{((1 + \eta\lambda)^K - 1)(1 - (\eta\lambda)^2)^K}.$$

$$(94)$$

*Proof.* For simplicity, we use $G(w) = w - \eta\nabla F(w)$ to represent the output of gradient descent of function $F(w)$. Similarly, we use $G_i(w) = w - \eta\nabla f_i(w)$ to denote the gradient descent output of the $i$-th individual loss function $f_i(w)$.

**Lemma 8** (Hardt et al. (2016)). *If $F(w)$ is convex and $\beta$-smooth, and $\eta \leq 2/\beta$, then the operation $G(w)$ is contractive, i.e.,*

$$\|G(w) - G(w')\| \leq \|w - w'\|,$$

*for arbitrary $w$ and $w'$. In addition, if $F(w)$ is $\lambda$-strongly convex and $\beta$-smooth, then if $\eta \leq 2(\beta + \lambda)$, then $G(w)$ is strictly contractive such that*

$$\|G(w) - G(w')\| \leq (1 - \frac{\eta\beta\lambda}{\beta + \lambda})\|w - w'\|.$$

In the $t$-th phase of Algorithm 1, conditional on the initialization $\bar{w}^{(t-1)}$, we first consider a virtual trajectory produced by applying full gradient descent on $F(w)$ with step size $\eta$ for $K$ iterations. We denote those iterates by $\tilde{w}^{(t,k)}$, for $k = 1, 2, \cdots, K$. Let $w^* = \arg\min_{w \in \mathcal{W}} F(w)$ be the global optimum, when $\eta < 1/\beta$,

$$\|\tilde{w}^{(t,k)}) - w^*\|^2 = \|\tilde{w}^{(t,k-1)} - w^* - \eta \nabla F(\tilde{w}^{(t,k-1)})\|^2 \tag{95}$$

$$\leq \|\tilde{w}^{(t,k-1)} - w^*\|^2 + \eta^2 \|\nabla F(\tilde{w}^{(t,k-1)})\|^2 - 2\eta(F(\tilde{w}^{(t,k-1)}) - F(w^*)) \tag{96}$$

$$\leq (1 - \eta\lambda)\|\tilde{w}^{(t,k-1)} - w^*\|^2 + (2\eta^2\beta - 2\eta)(F(\tilde{w}^{(t,k-1)}) - F(w^*)) \tag{97}$$

$$\leq (1 - \eta\lambda)\|\tilde{w}^{(t,k-1)} - w^*\|^2. \tag{98}$$

In (96), we use the property of strong convexity that

$$F(\tilde{w}^{(t,k-1)}) - F(w^*) \leq \langle \nabla F(\tilde{w}^{(t,k-1)}), \tilde{w}^{(t,k-1)} - w^* \rangle - \frac{\lambda}{2}\|\tilde{w}^{(t,k-1)} - w^*\|^2.$$

In (97), we use the smooth assumption that $\frac{1}{2\beta} \cdot \|\nabla F(\tilde{w}^{(t,k-1)})\|^2 \leq F(\tilde{w}^{(t,k-1)}) - F(w^*)$. Finally, in (98), as $\eta < 1/\beta$ and thus $2\eta(\eta\beta - 1) < 0$. Therefore,

$$\|\tilde{w}^{(t,K)} - w^*\|^2 \leq (1 - \eta\lambda)^K \|\bar{w}^{(t-1)} - w^*\|^2. \tag{99}$$

We will use $\gamma_1 = (1 - \eta\lambda)^K$ for simplicity.

Now, we consider to bound the deviation between $\tilde{w}^{(t,K)}$ and $\bar{w}^{(t)}$. In the following, we always assume $\eta < \min\{1/\beta, 2/(\beta + \lambda)\}$. It is noted that, based on the strict contraction property of $G$ and $G_i$, for any $u$ and $v$,

$$\|G_i(u) - G(v)\| = \|G_i(u) - G_i(v) + G_i(v) - G(v)\| \leq \|G_i(u) - G_i(v)\| + \|G_i(v) - G(v)\|$$

$$\leq (1 - \frac{\eta\beta\lambda}{\beta + \lambda})\|u - v\| + \eta\|\nabla f_i(v) - \nabla F(v)\|.$$

In the following, we use $\gamma_2 = (1 - \frac{\eta\beta\lambda}{\beta + \lambda})$ for simplicity. Similarly, for $\{G_1, G_2, \cdots, G_n\}$ on inputs $\{u_1, u_2, \cdots, u_n\}$, we have

$$\|\frac{\sum_{i=1}^n G_i(u_i)}{n} - G(v)\| \leq \gamma_2 \cdot \frac{\sum_{i=1}^n \|u_i - v\|}{n} + \|\frac{\sum_{i=1}^n G_i(v)}{n} - G(v)\|$$

$$= \gamma_2 \cdot \frac{\sum_{i=1}^n \|u_i - v\|}{n}. \tag{100}$$

At the $t$-th phase, from the initialization $\bar{w}^{(t-1)}$, $w_i^{(t,K)} = \underbrace{G_i \circ G_i \circ \cdots \circ G_i}_{k}(\bar{w}^{(t-1)})$. On the other hand, with the same start point $\bar{w}^{(t-1)}$, the virtual iterate $\tilde{w}^{(t,K)} = \underbrace{G \circ G \circ \cdots \circ G}_{k}(\bar{w}^{(t-1)})$.

Therefore, with a recursion reasoning,

$$\|\tilde{w}^{(t,K)} - \frac{\sum_{i=1}^n w_i^{(t,K)}}{n}\|$$

$$\leq \frac{\gamma_2 \cdot \sum_{i=1}^n \|w_i^{(t,K-1)} - \tilde{w}^{(t,K-1)}\|}{n}$$

$$\leq \frac{\gamma_2 \cdot \sum_{i=1}^n (\gamma_2 \|w_i^{(t,K-2)} - \tilde{w}^{(t,K-2)}\| + \eta\|\nabla f_i(\tilde{w}^{(t,K-1)}) - \nabla F(\tilde{w}^{(t,K-1)})\|)}{n} \tag{101}$$

$$\leq \|\bar{w}^{(t-1)} - \bar{w}^{(t-1)}\| + \frac{\eta \sum_{k=0}^{K-2} \gamma_2^{K-k} \sum_{i=1}^n \|\nabla f_i(\tilde{w}^{(t,k)}) - \nabla F(\tilde{w}^{(t,k)})\|}{n}$$

$$\leq \frac{\eta\sqrt{\tau}(1 - \gamma_2^K)}{1 - \gamma_2}.$$

Here, in (101), we apply Assumption 1 on the variance bound $\tau$, where the sampling noise of stochastic gradient satisfies $\|\sum_{i=1}^n (\nabla f_i(w) - \nabla F(w))\| \leq n\mathcal{B}$. Now, we further take the clipping

operation, i.i.d. sampling and DP noise into accountant. First, due to the clipping, stemmed from (101),

$$\|\frac{\sum_{i=1}^{n} \bar{w}^{(t-1)} + \mathcal{CP}(\Delta w_i^{(t)}, c)}{n} - \tilde{w}^{(t,K)}\| = \|\frac{\sum_{i=1}^{n} \bar{w}^{(t-1)} + \mathcal{CP}(w_i^{(t,K)} - \bar{w}^{(t-1)}, c)}{n} - \tilde{w}^{(t,K)}\|$$

$$\leq \|\frac{\sum_{i=1}^{n} \left(\bar{w}^{(t-1)} + \mathcal{CP}(w_i^{(t,K)} - \bar{w}^{(t-1)}, c) - w_i^{(t,K)}\right)}{n}\| + \|\frac{\sum_{i=1}^{n} w_i^{(t,K)}}{n} - \tilde{w}^{(t,K)}\|)$$

$$\leq \mathcal{B} + \frac{\eta\sqrt{\tau}(1 - \gamma_2^K)}{1 - \gamma_2}. \tag{102}$$

In the following, we proceed to incorporate the sampling noise and DP noise into the deviation analysis. Let $\mu^{(t)} = \frac{\sum_{i=1}^{n} \mathcal{CP}(\Delta w_i^{(t)}, c)}{n}$ be the average of clipped local update at the $t$-th phase. Let $\mathbf{1}_i^{(t)}$ to be an indicator which equals 1 iff the $i$-th sample gets selected (independently with rate $q$). Then,

$$\mathbb{E}[\|\bar{w}^{(t)} - \tilde{w}^{(t,K)}\|] = \mathbb{E}[\|\bar{w}^{(t-1)} + \frac{\sum_{i=1} \mathbf{1}_i^{(t)} \cdot \mathcal{CP}(\Delta w_i^{(t)}, c)}{nq} + e^{(t)} - \tilde{w}^{(t,K)}\|] \tag{103}$$

$$\leq \mathbb{E}[\|\bar{w}^{(t-1)} + \frac{\sum_{i=1} \mathbf{1}_i^{(t)} \cdot \mathcal{CP}(\Delta w_i^{(t)}, c)}{nq} - \tilde{w}^{(t,K)}\|] + \sigma\sqrt{d} \tag{104}$$

$$= \mathbb{E}[\|\bar{w}^{(t-1)} + \frac{\sum_{i=1} \mathbf{1}_i^{(t)} \cdot \mathcal{CP}(\Delta w_i^{(t)}, c)}{nq} - \mu^{(t)} + \mu^{(t)} - \tilde{w}^{(t,K)}\|] + \sigma\sqrt{d} \tag{105}$$

$$\leq \mathbb{E}[\|\frac{\sum_{i=1}(\mathbf{1}_i^{(t)} - q) \cdot \mathcal{CP}(\Delta w_i^{(t)}, c)}{nq}\| + \|\bar{w}^{(t-1)} - \tilde{w}^{(t,K)} + \mu^{(t)}\|] + \sigma\sqrt{d} \tag{106}$$

$$\leq \sqrt{\frac{nc^2}{n^2 q}} + \mathcal{B} + \frac{\eta\sqrt{\tau}(1 - \gamma_2^K)}{1 - \gamma_2} + \sigma\sqrt{d}. \tag{107}$$

In (104), we use the fact that $Q^{(t)}$ is independent DP noise with zero mean and $\mathbb{E}[\|Q^{(t)}\|] = \sigma\sqrt{d}$. In (106), we use the triangle inequality. In (107), we use the convexity of $l_2$ norm function and it is noted that $(\mathbf{1}_i^{(t)} - q)$ for $i = 1, 2, \cdots, n$, are i.i.d. and of zero mean while $\|\mathcal{CP}(\Delta w_i^{(t)}, c)\| \leq c$.

So far, we have derived the expected deviation between $\bar{w}^{(t)}$ and $\tilde{w}^{(t,K)}$ at the end of the $t$-th phase conditional on $\bar{w}^{(t-1)}$. In the following, we will continue to incorporate such deviation to (99).

By applying the AM-GM inequality, $\|u - v\|^2 \leq (1 + z)\|u\|^2 + (1 + \frac{1}{z})\|v\|^2$ for any $z > 0$, on $\|\bar{w}^{(t)} - w^*\|^2 = \|(\tilde{w}^{(t,K)} - w^*) + (\bar{w}^{(t)} - \tilde{w}^{(t,K)})\|^2$, we have that

$$\mathbb{E}[\|\bar{w}^{(t)} - w^*\|^2] \leq (1 + z)\mathbb{E}[\|\tilde{w}^{(t,K)} - w^*\|^2] + (1 + \frac{1}{z})\|\bar{w}^{(t)} - \tilde{w}^{(t,K)}\|^2]$$

$$\leq (1 + z)\gamma_1\mathbb{E}[\|\bar{w}^{(t-1)} - w^*\|^2] + (1 + \frac{1}{z})(\frac{c}{\sqrt{nq}} + \mathcal{B} + \frac{\eta\sqrt{\tau}(1 - \gamma_2^K)}{1 - \gamma_2} + \sigma\sqrt{d})^2$$

$$\leq (1 + z)\gamma_1\mathbb{E}[\|\bar{w}^{(t-1)} - w^*\|^2] + 4(1 + \frac{1}{z})(\frac{c^2}{nq} + \mathcal{B}^2 + \frac{\eta^2\tau(1 - \gamma_2^K)^2}{(1 - \gamma_2)^2} + \sigma^2 d) \tag{108}$$

Based on (108) by recursion, we further obtain the following unconditional expectation

$$\mathbb{E}[\|\bar{w}^{(T)} - w^*\|^2] \leq ((1 + z)\gamma_1)^T\|\bar{w}^{(0)} - w^*\|^2 + \frac{4(1 + \frac{1}{z})}{1 - (1 + z)\gamma_1}(\frac{c^2}{nq} + \mathcal{B}^2 + \frac{\eta^2\tau^2(1 - \gamma_2^K)^2}{(1 - \gamma_2)^2} + \sigma^2 d)$$

$$\leq (1 - (\eta\lambda)^2)^{TK}\|\bar{w}^{(0)} - w^*\|^2 + \frac{4(1 + \eta\lambda)^K \cdot (\frac{c^2}{nq} + \mathcal{B}^2 + \eta^2\tau K^2 + \sigma^2 d)}{((1 + \eta\lambda)^K - 1)(1 - (\eta\lambda)^2)^K} \tag{109}$$

In (109), we select $z = (1 + \eta\lambda)^K - 1$, $\qquad\qquad\qquad\qquad\qquad\qquad\qquad\qquad\qquad\qquad\qquad\qquad$ $\square$

## F  PROOF OF THEOREM 2: UTILITY OF DP-LSGD IN NON-CONVEX OPTIMIZATION

To apply Theorem 4 on DP-LSGD, we may equivalently view the perturbation term $Q^{(t)}$ as formed by two parts. One is due to the local update clipping and the other is the DP noise added, denoted by $e^{(t)}$ in this proof. To be formal, $Q^{(t)}$ can be rewritten as follows,

$$
Q^{(t)} = \frac{\eta}{nq} \sum_{i \in S_t} \sum_{k=0}^{K-1} (1 - \frac{c}{\max\{\|\sum_{k=0}^{K-1} g_i^k\|, c\}}) g_i^k + e^{(t)}
$$

$$
= \underbrace{\frac{\eta}{nq} \sum_{i=1}^{n} \sum_{k=0}^{K-1} 1_i^{(t)} (1 - \frac{c}{\max\{\|\sum_{k=0}^{K-1} g_i^k\|, c\}}) g_i^k}_{(A)} + e^{(t)}.
\tag{110}
$$

In (110), term (A) corresponds to the correction term due to the clipping, where equivalently the learning rate of the local update from each sample is scaled by a factor determined by the norm $\|\sum_{k=0}^{K-1} g_i^k\|$. $e^{(t)}$ is the independent DP noise added in the $t$-th phase. Therefore, conditional on $\bar{w}^{(t-1)}$, the expectation of $\|Q^{(t)}\|^2$ is in the following form,

$$
\mathbb{E}[\|Q^{(t)}\|^2] = \frac{\mathbb{E}[\|\sum_{i=1}^{n} \sum_{k=0}^{K-1} 1_i^{(t)} \eta (1 - \frac{c}{\max\{\|\sum_{k=0}^{K-1} g_i^k\|, c\}}) g_i^k\|^2]}{(nq)^2} + \sigma^2 d
$$

$$
\leq \frac{\sum_{i=1}^{n} \mathbb{E}[\|\eta(1 - \frac{c}{\max\{\|\sum_{k=0}^{K-1} g_i^k\|, c\}}) \sum_{k=0}^{K-1} g_i^k\|^2]}{nq} + \sigma^2 d
\tag{111}
$$

$$
= \frac{\sum_{i=1}^{n} \mathbb{E}[(\Psi_i^{(t)})^2]}{nq} + \sigma^2 d = q\mathcal{B}^2 + \sigma^2 d.
$$

Recall Definition 4, in (111), $\Psi_i^{(t)}$ is the incremental norm of the local update by $i$-th sample in the $t$-th phase, i.e., $\max\{\|\eta \sum_{k=0}^{K-1} g_i^k\| - c, 0\}$. Now, plugging the form of $\mathbb{E}[\|Q^{(t)}\|^2]$ in (111) back to Theorem 4, we obtain the utility bound claimed for DP-LSGD.

## G  ADDITIONAL EXPERIMENTS AND EXPERIMENT SETUPS

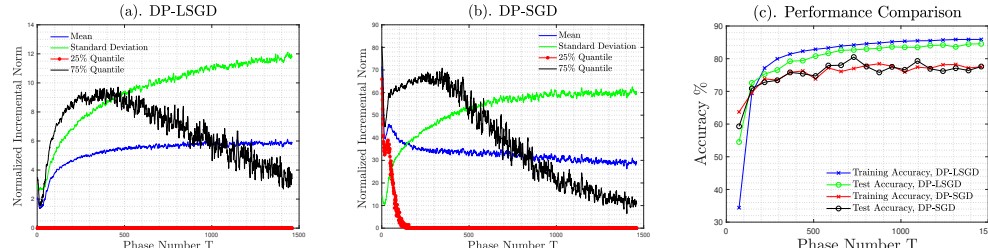

Figure 4: Training ResNet 20 on SVHN with DP-LSGD ($K = 10, \eta = 0.025, c = 1$) and DP-SGD ($K = 1, \eta = 1, c = 1$) under ($\epsilon = 2, \delta = 10^{-5}$)-DP, with expected batch size 1000.

For all the experiments with respect to CIFAR10, we assume the training data set of 50,000 samples is private. Similarly, for SVHN, we assume the training data set of 73,257 samples is private. In Fig. 4 (a,b), we report the statistics of normalized incremental norm when we train ResNet20 on SVHN. Very similar to our observation on CIFAR10, both the mean and the standard deviation of the normalized incremental norm in DP-LSGD is only about a half of those in DP-SGD, which suggest that DP-LSGD bears less influence from the clipping operator. As a consequence, in Fig. 4 (c), we can see DP-LSGD enjoys a faster convergence rate accompanying with a better utility-privacy tradeoff.

As for the hyper-parameter selection, in Table 1, we first fixed the clipping threshold $c = 1$ and conducted grid searches on the learning rate $\eta \in \{0.125, 0.25, 0.5, 1, 2, 4\}$ and composition budget

| Hyper $\setminus$ $\epsilon$ | 1.0 | 1.5 | 2.0 | 2.5 | 3.0 | 4.0 |
|---|---|---|---|---|---|---|
| Step Size $\eta$ | 0.5 | 1 | 1 | 1 | 2 | 2 |
| Composition Budget $T$ | 500 | 1000 | 1000 | 1500 | 1500 | 2000 |

Table 3: Optimal Hyper-parameter Selection of DP-SGD

$T \in \{500, 1000, 1500, 2000, 2500\}$ for DP-SGD in various $(\epsilon, \delta)$-DP setups, where empirically the optimal selection is shown in Table 3.

Provided the optimal hyperparameter setup of DP-SGD, for DP-LSGD we also fixed $c = 1$ and adopted the same composition budget $T$ as selected in Table 3 and moved on to optimize the step size $\eta \in \{0.0125, 0.025, 0.05, 0.1\}$ and local iteration number $K \in \{5, 10, 15, 20\}$. We found $K = 10$ and $\eta = 0.025$ consistently being the optimal selection in all cases, as summarized in Table 4.

| Hyper $\setminus$ $\epsilon$ | 1.0 | 1.5 | 2.0 | 2.5 | 3.0 | 4.0 |
|---|---|---|---|---|---|---|
| Step Size $\eta$ | 0.025 | 0.025 | 0.025 | 0.025 | 0.025 | 0.025 |
| Composition Budget $T$ | 500 | 1000 | 1000 | 1500 | 1500 | 2000 |
| Local Iteration $K$ | 10 | 10 | 10 | 10 | 10 | 10 |

Table 4: Optimal Hyper-parameter Selection of DP-LSGD

In Table 2 with comparisons to De et al. (2022), for DP-SGD on WideResNet-40-4, we adopted the same parameter $c$, $T$ and $\eta$ suggested in Appendix C.5 of De et al. (2022), while for DP-LSGD we applied the same $c$ and $T$ while similarly selected $K = 10$ and $\eta = 0.025$.

