# OpenReview forum: "Why DP "LOCAL" SGD – Faster Convergence in Less Composition with Clipping Bias Reduction"
_ICLR.cc/2025/Conference — ICLR 2025 Conference Withdrawn Submission_

### Official Review · Reviewer_AJpB · 2024-11-02

**Soundness:** 2
**Presentation:** 2
**Contribution:** 2
**Rating:** 6
**Confidence:** 3

**Summary:**

This paper proposes a modification to the popular DP-SGD algorithm by taking multiple local steps for each sample. Instead of aggregating the per-example gradients under DP, the local updates after multiple steps are aggregated with the Gaussian mechanism. Authors derive theoretical convergence results demonstrating that the proposed method can achieve faster convergence than the basic DP-SGD in both convex and non-convex settings. The proposed method is compared against DP-SGD in learning ResNet20 and a WRN-40-4 models with various data sets, with the proposed method outperforming DP-SGD for all the data sets and all privacy levels considered.

**Strengths:**

The utility analysis for the algorithm, especially the non-convex case, is a very interesting result. Since DP-SGD would correspond a special case (K=1) of the algorithm, authors can easily compare the DP-SGD against DP-LSGD in Theorems 1 and 2. Furthermore, the convergence result for the non-convex case is obtained without Lipschitzity of the loss function which makes the proposed result more appealing than the existing ones that operate under more limited set of losses.

The empirical results seem very impressive. The proposed algorithm can outperform DP-SGD in every experiment, with even a rather substantial margin in the low epsilon regime.

**Weaknesses:**

To me, the proposed algorithm is essentially the same as DP-FedAvg (McMahan et al. 2018), where the clients of an FL setting have been replaced with individual samples. Therefore the novelty in the algorithm is somewhat limited. However, I'm not aware of any earlier study that would have applied DP-FedAvg in this setting.

The writing could be improved, especially in the theoretical section. I will list some questions regarding these clarity issues in the next section.

McMahan et al 2018. Learning differentially private recurrent language models. ICLR 2018

**Questions:**

- On page 6 you state that you focus on a practical scenario where $\mathcal{B} = O(c)$. Can you clarify why that is practical? From Def. 4, I would imagine the $\Psi$s can be arbitrarily large for the clipped updates, and hence I don't see how the $\mathcal{B}$ would be bounded in $O(c)$.
- When discussing the convex case (Thm. 1), you mention that the DP-LSGD, with suitable choices of hyperparameters, can enjoy $O(1/T)$ convergence rate while DP-SGD has $O(1/\sqrt{T})$ rate. However, the DP-LSGD requires computing $K = O(T)$ local steps to achieve this, which seems to suggest that there are in total $O(T^2)$ update steps (local steps $\times$ global steps). Maybe I'm wrong, but this would seem to imply that the number of computation is not comparable between the DP-SGD and DP-LSGD.
    - Could you run some of the experiments by allowing DP-SGD to run for $KT$ iterations instead of $T$ iterations? E.g. in Fig 1 you could run the DP-SGD for 15000 iterations and check if the increased iteration budget helps with the convergence.
- You mention "In Fig. 1 (c) ... which aligns with our theory that DP-LSGD exhibits a smaller clipping bias ...": how can you assess the clipping bias from this result? I can buy the faster convergence argument, but I think the clipping bias would require some further studies.
- I guess the $B$ e.g. in Fig 2 should be $\mathcal{B}$?
- It would be beneficial to have some error bars in Table 2, to understand the statistical significance of the improved accuracy of DP-LSGD.

---

### Official Review · Reviewer_fE5L · 2024-11-02

**Soundness:** 2
**Presentation:** 2
**Contribution:** 2
**Rating:** 3
**Confidence:** 4

**Summary:**

The authors analyze DP-LSGD as an alternative to DP-SGD and provide theorectical analysis of the convergence of it. Furthermore, the work provides experimental evidence displaying the difference between the performance of DP-SGD and DP-LSGD on image classification datasets.

**Strengths:**

- The authors provide theorectical analysis on the convergence of DP-LSGD.

- The authors execute experiments on multiple image classification benchmarks.

- The authors provided the source code which allowed to understand the actual implementation of the method and details better.

**Weaknesses:**

I focus my analysis mostly on the experimental results and the computational complexity as I believe there are major problems in these sections that lead me at the moment of writing the review (without having the expertise of the other reviewers) to **not** recommending the acceptance of the paper.

- **W1: Batch sampling and accounting do not match.** (Short version, details below): The authors do the accounting based on Poisson subsampling but the batches are not generated using Poisson subsampling but using a normal `torch.utils.data.DataLoader`. This is unlike implemented in other DP-SGD frameworks like opacus [1] and has been shown to possibly result in different $(\epsilon, \delta)$-DP guarantees [2].

- **W2: Comparison to De et al..** The authors compare to De et al. [5] in Table 2 and write that the method "outperforms De et al. (2022)". This is also a big chunck of their contribution 3. There are multiple issues with this claim: (1): The authors do not provide error bars and the gap between their method and De et al. is not big. The authors could make their claim substantial by running multiple repeats of both methods to make clear if one of them is better. (2) The numbers reported by the authors in Table 2 are worse than what is reported by De et al. in their Table 3 [5], e.g. at $\epsilon=8$ the authors report for their method accuracy of 80.6 and for De et al. 80.3 while the original Table 3 of De et al. reports $81.4\pm0.2$. The authors could clarify why they report weaker accuracy for De et al. [5], report the actual numbers of De et al. or rerun the experiments in a way that makes clear why their numbers are lower. In the current form this comparison is misleading and the numbers in the contribution 3 are wrong.

- **W3: Accounting when comparing with others.** The authors implement their accounting based on the `prv_accountant` [3] whereas De et al. [5] use the `RDP accountant`. The `RDP accountant` leads usually to higher noise_multiplier $\sigma$ than PRV at the same $(\epsilon, \delta)$ given subsampling rate $q$ and iterations $T$. Given the weakness W3 it is not clear to me if any additional reported gap between the authors and De et al. [2] could be explained by this unjust comparison. The authors could improve on this by comparing methods using the same accounting and furthermore mentioning what their accounting is based on in the text. I did not manage to find any mention in the paper regarding that and had to resort to the source code to find that out.

- **W4: Computational Complexity and Runtime.** The authors write in Section 5 that a possible limitation of DP-LSGD that the runtime is $k$ times higher than with DP-SGD. I think this is a limitation that needs to be mentioned more prominently. Usually, training longer (e.g., for more steps $T$ or epochs with a higher noise_multiplier) or with a larger subsampling_ratio $q$ is beneficial but the constraint is the amount of compute available. See a discussion of this in Section 5.4.1 of Ponomareva et al. [6]. To understand better what the actual benefit of DP-LSGD is, I would suggest experiments that illustrate the performance when the available compute is fixed as at the moment the authors compare in a way that gives DP-LSGD a magnitude more compute resources ($k=10$ in many experiments).


Details on W1:
- **Details for Batch sampling and accounting do not match** The authors use the function `get_std` in line 145 to determine the noise_multiplier $\sigma$. This is subsequently done using the class `Accountant` of the `prv_accountant` [3] which relies on Poisson subsampling [4]. But instead of implementing poisson subsampling something else is implemented: In line 119-123 of `model_local.py` a normal `torch.utils.data.DataLoader` with `shuffle=True` is used and subsequently in line 160 just used as it is. This is unlike implemented in other DP-SGD frameworks like opacus [1] and has been shown to possibly result in different $(\epsilon, \delta)$-DP guarantees [2]. I believe what the authors have implemented is something like Algorithm 3 of [2] with the severity of the problem depending on the exact subsampling rate $q$, iterations $T$ and noise_multiplier $\sigma$. I did not dig deeper into the code and computed the actual $(\epsilon, \delta)$-DP gurantees for the implementation of the authors. Interestingly, the authors speak about (expected) batch size throughout the experimental section in their paper what might imply some randomness, but the sampled batch_size is constant in the implementation.

Minor:
- **The noise_multiplier could be lower than expected in `get_std`**: You set the noise_multiplier $\sigma = 100$ whenever some exception occurs. I believe this is quite dangerous as the required $\sigma$ might need to be higher than $100$. I would recommend you to not handle your exceptions like this as this might be the source of some silent bugs. I didn't check if this an effect on your experiments.



[1] Opacus DPDataloader: https://github.com/pytorch/opacus/blob/main/opacus/data_loader.py

[2] Chua, L., Ghazi, B., Kamath, P., Kumar, R., Manurangsi, P., Sinha, A., & Zhang, C. How Private are DP-SGD Implementations?. In ICML 2024.

[3] Gopi, S., Lee, Y. T., & Wutschitz, L. (2021). Numerical composition of differential privacy. In NeurIPS 2021.

[4] prv_accountant Accountant class: https://github.com/microsoft/prv_accountant/blob/a844f7200af2722d6a81e0386b2b7351688ecf35/prv_accountant/accountant.py#L145

[5] De, S., Berrada, L., Hayes, J., Smith, S. L., & Balle, B. (2022). Unlocking high-accuracy differentially private image classification through scale.arXiv:2204.13650.

[6] Ponomareva, N., Hazimeh, H., Kurakin, A., Xu, Z., Denison, C., McMahan, H. B., ... & Thakurta, A. G. (2023). How to dp-fy ml: A practical guide to machine learning with differential privacy. JAIR, 77, 1113-1201.

**Questions:**

- Q1: Did I understand the code correctly and do you really not use Poisson subsampling? (Re W1)
- Q2: Why are the numbers from De et al. so lower in your paper than in the original paper? (Re W2)
- Q3: Could you please clarify the accounting and how it differs between the comparisons that you make? (Re W3)
- Q4: Could you elaborate on the run-times and why you think having a magnitude more compute is a fair comparison? (Re W4)

---

### Official Review · Reviewer_u2zs · 2024-11-03

**Soundness:** 3
**Presentation:** 2
**Contribution:** 3
**Rating:** 5
**Confidence:** 3

**Summary:**

This research presents DP-LSGD, a new and generalized version of the differential privacy stochastic gradient descent (DP-SGD) algorithm. Inspired by the LSGD algorithm used in federated learning to minimize communication costs, DP-LSGD allows each example in a batch to perform multiple local model updates in parallel. These multi-step updates are then clipped, averaged, and privatized before being applied to the global model.  Essentially, DP-SGD is simply a special case of DP-LSGD with only one local update. Importantly, for the same clipping threshold and number of global updates, the privacy guarantees of DP-LSGD match those of DP-SGD.

The authors provide a theoretical analysis of DP-LSGD's convergence in both convex and non-convex settings, assuming the gradient has a bounded second moment. Their findings demonstrate that with an appropriate number of local updates, DP-LSGD converges significantly faster than DP-SGD. While the paper outlines the proof for the convex case, detailed proofs for both settings, along with additional results, are included in the appendix.

Empirical evaluations further validate the effectiveness of DP-LSGD. The authors present statistics on clipping bias for both DP-SGD and DP-LSGD, and compare the performance of the two algorithms in training ResNet20 models on CIFAR10, SVHN, and EMNIST datasets across various epsilon values. Their results consistently show that DP-LSGD with 10 local updates surpasses DP-SGD in performance.

The authors acknowledge that DP-LSGD requires significant memory to store per-example gradients and local model updates and suggest optimizations as future work.

**Strengths:**

- Algorithm uses ideas from federated learning to provide better theoretical and empirical performance compared to DP-SGD.
- Experimental evaluations are conducted on multiple datasets and confidence intervals are provided for most experiments.

**Weaknesses:**

- Experimental evaluation is not very clear. It is unclear why K = 10 always performs the best? Also, how does the choice of clipping threshold impact K?
- Would it be better to compare DP-SGD with only the privacy requirement and optimize all other hyperparameters including clipping threshold and number of iterations?

**Questions:**

- Could this be practical for very large models where per core batch sizes are very small or 1?
- In many applications, I think it has been seen empirically that it works well when clipping threshold is very small and almost all gradients are clipped. In that scenario, I am not sure if local updates provides any benefit. How was the clipping threshold chosen here?

---

### Official Review · Reviewer_PgWH · 2024-11-04

**Soundness:** 3
**Presentation:** 2
**Contribution:** 3
**Rating:** 3
**Confidence:** 3

**Summary:**

The paper considers the local DP-SGD algorithm, i.e. DP-SGD but we perform a few steps of local gradient descent on each loss function and report the sum of the updates as the object to be clipped / noised, rather than a single gradient update. They show convergence bounds for the last iterate (which is more challenging than the usual average iterate analysis) of local DP-SGD for convex smooth losses and non-convex smooth losses, without a Lipschitz assumption but rather a bounded variance assumption / a bounded variance on what the authors call the incremental norm, i.e. a bound on the average per-example update bias due to clipping. Motivated by this, the authors conduct experimental results where they study the bias due to clipping, and show that local DP-SGD outperforms DP-SGD on CIFAR10 training under the same privacy budget.

**Strengths:**

* The problem studied of understanding the effects of clipping on model training is an important yet challenging problem, especially for Local SGD which is strictly more general than SGD.
* The empirical results are strong and go beyond just comparing accuracies, instead studying quantities like the incremental norm in detail to help validate the authors' intuition for the behavior of Local SGD developed in the paper. Even just focusing on the accuracy, while Local SGD is a well-known algorithm, these results suggest it should be given further consideration in settings where DP-SGD is the default algorithm.

**Weaknesses:**

* The main weakness of the paper is that there is a lack of clarity / consistency on the settings of several parameters in the theoretical results, or some of the terms that arise in the bounds. See Questions below for more details.  I understand the authors are using assumptions that are not common in the literature as they are needed to answer the new theoretical questions the authors study, so comparisons to past work to help contextualize the authors' results may generally be tricky. Nevertheless, I would suggest the authors more carefully discuss the terms in their bounds and specific parameter choices to avoid some of the confusion in the questions mentioned below, and to give a reader more confidence in the improvements claimed in the discussions of the main theorems. So even though the results seem like they could be quite impressive and resolve some challenging and important research problems, it is hard for me to suggest accepting this paper because it's hard to interpret the theoretical results in the paper or how meaningful they are.

**Questions:**

* In the discussion following Theorem 1, the authors say to consider a practical scenario where $B = O(c)$. If $B = \Theta(c)$, the clipping bias term is roughly $\gamma$, the maximum difference between two functions. Any point in the convex hull of the per-example minimizers must have excess loss at most $\gamma$ by convexity, and (ignoring DP noise) SGD with clipping will converge to this convex hull. This seems to suggest if one is paying a factor of $\gamma$ one can just analyze the clipped losses and avoid the need to understand the impact of clipping. In this case why does one need a bound on $B$? Or, should we think of $B = o(c)$ instead? If so, what is the justification for this? In short, if we should really consider $B = \Theta(c)$, I think the term $\gamma$ needs a more involved discussion, and some justification for why understanding the impact of clipping bias is interesting/necessary if our bound is lower bounded anyway by $\gamma$.
* In Theorem 1 the authors say to set $K^2 = O(nq)$, but at the bottom of page 6 give another parameter setting for $K$ of $O(T d / n^2 \epsilon^2)$. Under this setting, for small dimensions it seems like $q$ might be $o(1/n)$ (i.e. you take less than one example per iteration) unless one chooses $T = \Omega(n^2)$ which seems excessive. How does one reconcile these two different choices?
* In Theorem 2, why does the privacy term $d^2 / \epsilon^2 n^2$ not grow with $T$ while it does in Theorem 1? In general for DP without some strong-convexity parameter, I would expect this term to increase in $T$ as the noise should go to infinity as $T$ goes to infinity (if other parameters are fixed).
* In the discussion following Theorem 2, why is $B = B_0 \cdot \eta$ a practical scenario? Since your choice of $\eta$ is decaying in $K$ and $T$ this means $B$ is also decaying in these parameters, but as I understand it $B$ seems like it should be independent of $T$ and maybe increasing in $K$ since more steps of local SGD on non-convex losses will cause the local updates to have higher variance. In particular, this assumption on $B$ seems to lead to the issue in the previous bullet, where the bound under this assumption isn't paying any cost in privacy for more iterations.

---

### Note · Authors · 2024-11-12

I have read and agree with the venue's withdrawal policy on behalf of myself and my co-authors.